# The stress-responsive kinases MAPKAPK2/MAPKAPK3 activate starvation-induced autophagy through Beclin 1 phosphorylation

Yongjie Wei[1,2†], Zhenyi An[1†], Zhongju Zou[1,2], Rhea Sumpter Jr[1], Minfei Su[3], Xiao Zang[4], Sangita Sinha[3], Matthias Gaestel[5], Beth Levine[1,2,6*]

[1]Center for Autophagy Research, Department of Internal Medicine, UT Southwestern Medical Center, Dallas, United States; [2]Howard Hughes Medical Institute, UT Southwestern Medical Center, Dallas, United States; [3]Department of Chemistry and Biochemistry, North Dakota State University, Fargo, United States; [4]Department of Clinical Sciences, UT Southwestern Medical Center, Dallas, United States; [5]Institute of Physiological Chemistry, Hannover Medical School, Hannover, Germany; [6]Department of Microbiology, UT Southwestern Medical Center, Dallas, United States

**Abstract** Autophagy is a fundamental adaptive response to amino acid starvation orchestrated by conserved gene products, the autophagy (ATG) proteins. However, the cellular cues that activate the function of ATG proteins during amino acid starvation are incompletely understood. Here we show that two related stress-responsive kinases, members of the p38 mitogen-activated protein kinase (MAPK) signaling pathway MAPKAPK2 (MK2) and MAPKAPK3 (MK3), positively regulate starvation-induced autophagy by phosphorylating an essential ATG protein, Beclin 1, at serine 90, and that this phosphorylation site is essential for the tumor suppressor function of Beclin 1. Moreover, MK2/MK3-dependent Beclin 1 phosphorylation (and starvation-induced autophagy) is blocked in vitro and in vivo by BCL2, a negative regulator of Beclin 1. Together, these findings reveal MK2/MK3 as crucial stress-responsive kinases that promote autophagy through Beclin 1 S90 phosphorylation, and identify the blockade of MK2/3-dependent Beclin 1 S90 phosphorylation as a mechanism by which BCL2 inhibits the autophagy function of Beclin 1.

*For correspondence: beth. levine@utsouthwestern.edu

†These authors contributed equally to this work

## Introduction

When eukaryotic cells are deprived of amino acids, they activate autophagy, a conserved lysosomal degradation pathway that enables cells to survive short-term periods of starvation by recycling nutrients and maintaining energy homeostasis (*Levine and Klionsky, 2004*; *Mizushima and Komatsu, 2011*). The autophagy pathway is mediated by a set of evolutionary conserved proteins, the ATG proteins, that include proteins involved in the Atg1/ULK1 serine/threonine kinase protein complex, the Beclin 1/VPS34 class III phosphatidylinositol kinase (PI3K) complex, and the ATG5/ATG12/ATG16 and Atg8/LC3 protein conjugation complexes (*Levine and Klionsky, 2004*). The activity of these ATG protein complexes can be regulated by protein–protein interactions, post-translational modifications (e.g., ubiquitination, phosphorylation, acetylation), and at the transcriptional level (*Kroemer et al., 2010*; *Abrahamsen et al., 2012*; *Norris and Yao, 2012*; *Bánréti et al., 2013*; *Webster et al., 2014*).

Despite extensive research over the past several years into the molecular mechanisms of autophagy, very little is understood about stress-responsive signals that regulate activation of

**eLife digest** Cells keep themselves healthy by breaking down unneeded or damaged internal structures via a process called autophagy. This process also helps a cell to survive if it is starved of nutrients. For example, if a cell does not receive enough amino acids, it cannot make new proteins. Autophagy can break down existing non-essential proteins so that their amino acids can be re-used to build other proteins that the cell needs to survive.

Autophagy is performed by a set of proteins that is found in many different species, ranging from yeast to humans and plants. How these proteins are activated when a cell is starved of amino acids is not fully understood. However, evidence suggests that activating one of these proteins, called Beclin 1, by adding phosphate groups to it controls the extent to which autophagy occurs. It is also known from previous work that less autophagy occurs when Beclin 1 binds to another protein called BCL2.

Wei, An et al. identified two enzymes that attach a phosphate group to a specific site on Beclin 1 to activate it, and revealed that autophagy is defective in cells that lack these enzymes. Furthermore, Wei, An et al. found the BCL2 protein prevents autophagy by binding to Beclin 1 in such a way that stops these two enzymes from activating Beclin 1.

Beclin 1 is also known to prevent the growth of malignant tumors. Wei, An et al. found that to do so, Beclin 1 must have a phosphate group added to the same site that activates the protein during autophagy. This suggests that drugs that enhance the addition of this phosphate group to Beclin 1 could help activate autophagy and have anti-cancer effects.

autophagy protein complexes during amino acid deprivation. Inactivation of the autophagy repressor kinase mTOR (mechanistic target of rapamycin) in response to amino acid starvation contributes to autophagy through phosphorylation of ULK1/2 and ATG13 (resulting in their dissociation) and of ATG14 (*Meijer et al., 2014*). The low-energy sensing kinase, AMPK, also acts directly on components of the autophagy machinery (ULK1, Beclin 1, VPS34) to positively regulate glucose starvation-induced autophagy (*Egan et al., 2011*; *Kim et al., 2013*), but does not appear to function similarly in amino acid starvation-induced autophagy (*Kim et al., 2013*). Thus, an important open question is whether any stress-responsive signals directly regulate autophagy proteins to promote autophagy during amino acid starvation.

The autophagy protein, Beclin 1, is a central regulator of autophagy and functions through its interaction with the Class III PI3K VPS34, VPS15, and the autophagy protein ATG14 in the initial stages of autophagosome formation (*Funderburk et al., 2010*; *He and Levine, 2010*; *Abrahamsen et al., 2012*; *Fu et al., 2013*). It is a haploinsufficient tumor suppressor important for breast cancer and also acts in various other biological processes, including differentiation and development, innate immunity, lifespan extension, protein against neurodegeneration, and exercise-mediated effects on glucose metabolism (*Cao and Klionsky, 2007*; *Levine and Kroemer, 2008*; *He et al., 2012*). During amino acid starvation, Beclin 1-associated VPS34 autophagy complex activity is negatively regulated by the direct binding of Beclin 1 to BCL2 or BCL2L1 (*Sinha and Levine, 2008*), and by inhibitory interactions with the oncogenic kinases, AKT and EGFR (*Wang et al., 2012*; *Wei et al., 2013*), the Golgi-associated protein, GLIPR2/GAPR-1 (*Shoji-Kawata et al., 2013*), and the kinase MST1 (*Maejima et al., 2013*). Several proteins positively regulate the Beclin 1-associated VPS34 autophagy complex, including those that disrupt BCL2/BCL2L1-Beclin 1 binding by either Beclin 1 phosphorylation (e.g., DAPK, ROCK [*Zalckvar et al., 2009*; *Gurkar et al., 2013*]) or BCL2 phosphorylation (e.g., JNK1 [*Wei et al., 2008*]) and proteins that are part of the core autophagy machinery, such as ULK1 and AMBRA1 (*Fimia et al., 2007*; *Di Bartolomeo et al., 2010*; *Russell et al., 2013*). These observations suggest that Beclin 1 may be a central target that is 'turned on' or 'turned off' in order to properly coordinate autophagosome formation with the needs of the cell to increase or decrease autophagy in a context-dependent manner.

However, it remains unknown whether stress-activated kinases directly phosphorylate Beclin 1 (or other ATG proteins) to positively regulate autophagy in response to amino acid starvation. Here we describe a novel role for members of the stress-activated protein kinase (p38 MAPK) signaling pathway, MAPKAPK2 (MK2) and MAPKAPK3 (MK3) (*Cargnello and Roux, 2011*), in mediating Beclin 1 S90 phosphorylation, which we show is essential for its autophagy and tumor

suppressor function. This MK2/3-dependent Beclin 1 S90 phosphorylation is inhibited by a mutant form of BCL2 that cannot be phosphorylated by the MAPK family member, JNK1.

Our findings reveal a crucial link between an evolutionarily conserved family of stress-activated protein kinases, MK2 and MK3, and a component of the autophagy machinery, Beclin 1, that plays an essential role in starvation-induced autophagy. Moreover, we demonstrate that blockade of MK2/MK3-dependent Beclin 1 phosphorylation is a mechanism by which BCL2 family proteins inhibit autophagy. These finding have important implications for understanding how cells respond to amino acid starvation and turn on autophagy.

## Results

### Nutrient starvation enhances Beclin 1 S90 phosphorylation

Activation of the Beclin 1/VPS34 complex is an essential event for starvation-induced autophagy (*Funderburk et al., 2010*). To gain insights into potential mechanisms that regulate starvation-induced autophagy, we sought to identify post-translational modifications of Beclin 1 that occur specifically in response to the autophagy-inducing condition, amino acid starvation. Mass spectrometry analysis of SKNSH human neuroblastoma cells stably transfected with Flag epitope-tagged Beclin 1 grown in normal growth or starvation conditions identified only one phosphorylation event that increased specifically in response to amino acid deprivation—phosphorylation of the serine 90 residue of Beclin 1 (data not shown).

We confirmed starvation-induced phosphorylation of Beclin 1 S90 using HeLa cells transiently transfected with either Flag epitope-tagged wild-type Beclin 1 or Flag epitope-tagged mutant Beclin 1 containing a non-phosphorylatable alanine substitution at residue 90 of human Beclin 1 (Beclin 1 S90A). Following metabolic labeling with $^{33}$P, immunoprecipitation with an anti-Flag antibody and autoradiography, minimal phosphorylation was detected during nutrient-rich conditions, whereas Beclin 1 phosphorylation increased in response to starvation (*Figure 1A*, upper gel). Mutation of S90 to alanine almost completely abolished the $^{33}$P signal of Beclin 1, indicating that S90 is a major starvation-induced phosphorylation site (*Figure 1A*, upper gel). We confirmed that starvation induces Beclin 1 S90 phosphorylation by generating a phosphospecific antibody that recognizes Beclin 1 p-S90. Beclin 1 p-S90 reactivity of wild-type Beclin 1, but not Beclin 1 S90A, increased after nutrient starvation (*Figure 1A*, middle gel).

We next evaluated whether endogenous Beclin 1 undergoes S90 phosphorylation in response to nutrient (amino acid) starvation using the Beclin 1 S90 phosphospecific antibody. Beclin 1 S90 phosphorylation was detected in HeLa cell lysates within 30 min after starvation and gradually increased over a 2 hr period (*Figure 1B*). This increase in Beclin 1 S90 phosphorylation was accompanied by starvation-induced autophagy as measured by a time-dependent decrease in levels of the autophagy substrate, SQSTMI/p62 (herein referred to as p62), and increase in the conversion of the autophagy protein LC3-I to the lipidated, autophagosome-associated form, LC3-II (*Figure 1B*). Thus, endogenous Beclin 1 undergoes phosphorylation at residue S90 in response to nutrient starvation in parallel with autophagy induction.

### The Beclin 1 S90 phosphorylation site is required for starvation-induced autophagy

Next, we asked whether Beclin 1 S90 phosphorylation is essential for starvation-induced autophagy. To evaluate this question, we used two different cell lines that are deficient in Beclin 1 expression and starvation-induced autophagy, including human MCF7 breast carcinoma cells (*Liang et al., 1999*) and U2OS cells that inducibly express shRNA targeted against *beclin 1* (*Sun et al., 2008*).

MCF7 cells were derived from a patient with allelic loss of *beclin 1*, have low levels of endogenous Beclin 1 expression, and are defective in starvation-induced autophagy in the absence of exogenous *beclin 1* gene transfer (*Liang et al., 1999*, *2001*; *Furuya et al., 2005*; *Pattingre et al., 2005*; *Wang et al., 2012*). As reported, enforced expression of wild-type Beclin 1 rescued starvation-induced autophagy, as measured by decreased levels of p62, increased LC3-II conversion and increased numbers of GFP-LC3 puncta (a marker for autophagosomes) (*Figure 2A–C*). These readouts represented an increase in autophagic flux rather than a block in autophagosomal maturation, as treatment with the lysosomal inhibitor bafilomycin A1 blocked p62 degradation and further increased LC3-II accumulation and numbers of GFP-LC3 puncta (*Figure 2B,C*). In contrast, enforced

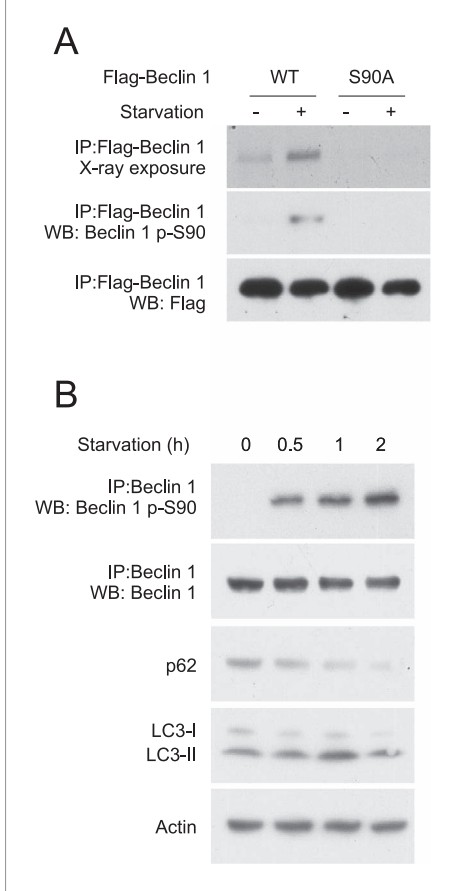

**Figure 1**. Beclin 1 S90 is phosphorylated in response to nutrient starvation. (**A**) Detection of Beclin 1 S90 phosphorylation in HeLa cells transfected with indicated Flag-Beclin 1 construct by radiolabeling, immunoprecipitation with an anti-Flag antibody and autoradiography (upper gel) or by western blot analysis using an anti-Beclin 1 S90 phosphospecific antibody (middle gel). Starvation (−) refers to growth in normal medium and starvation (+) refers to growth in HBSS for 2 hr. (**B**) Detection of endogenous Beclin 1 S90 phosphorylation in HeLa cells by immunoprecipitation with anti-Beclin 1 followed by immunoblot analysis with anti-Beclin 1 p-S90 phosphospecific antibody and detection of autophagy by p62 and LC3 immunoblot analysis in HeLa cells at indicated time points after starvation in HBSS. Actin is shown as a loading control.

expression of the Beclin 1 S90A mutant failed to induce autophagy in response to starvation (**Figure 2A–C**), indicating that the Beclin 1 S90 phosphorylation site is essential for autophagy induction in response to nutrient starvation. Moreover, a phosphomimetic mutant Beclin 1 S90E increased autophagy in basal conditions, suggesting that Beclin 1 S90 phosphorylation may be sufficient to induce autophagy (**Figure 2A–C**).

We observed similar results in U2OS cells with doxycycline inducible shRNA knockdown of endogenous Beclin 1. Doxycycline treatment of these cells resulted in undetectable levels of Beclin 1 and lack of starvation-induced p62 degradation. Consistent with previous reports of Beclin 1 knockdown or knockout in other mammalian cells (**Matsui et al., 2007**; **He et al., 2013**; **Mandell et al., 2014**) and knockout of Atg6 in yeast (**Suzuki et al., 2004**), knockdown of Beclin 1 did not block LC3 lipidation (**Figure 2—figure supplement 1**) but it did block the formation of GFP-LC3 puncta (**Figure 2D**, data not shown) (Atg6/Beclin 1 are not invariably required for LC3 lipidation, but they are required for the localization of lipidated LC3 to the autophagosome [**Mizushima et al., 2010**]). Expression of shRNA-resistant wild-type Beclin 1, but not shRNA-resistant Beclin 1 S90A, rescued starvation-induced autophagic flux as measured by p62 degradation and quantification of GFP-LC3 puncta in the presence and absence of bafilomycin A1 (**Figure 2D,E**). Expression of the phosphomimetic mutant Beclin 1 S90E increased autophagy in basal conditions to levels similar to those observed in starvation in cells expressing wild-type Beclin 1 (**Figure 2D,E**). Taken together, the data in MCF7 cells and U2OS cells provide strong evidence that Beclin 1 S90 phosphorylation is both necessary and sufficient for autophagy induction. We note that there is not a complete block in autophagic flux in empty vector transfected MCF7 cells or in U2OS cells treated with doxycycline, presumably due to the presence of low levels of Beclin 1 expression.

In both MCF7 cells and U2OS cells, we observed that bafilomycin A1 treatment resulted in Flag-Beclin 1 S90 phosphorylation in the absence of starvation. To determine whether bafilomycin A1 also induces phosphorylation of endogenous Beclin 1 S90 and whether such effects are due to reactive oxygen species (ROS) generation that occurs as a consequence of inhibition of vacuolar-type-H⁺-ATPases (**Zhdanov et al., 2011**; **Yokomakura et al., 2012**), we measured the effects of bafilomycin A1 treatment on Beclin 1 S90 phosphorylation in HeLa cells in the presence or absence of the ROS scavenger, *N*-acetyl-L-cysteine (**Figure 2—figure supplement 2**). Our results indicate that bafilomycin A1 results in endogenous Beclin 1 S90 phosphorylation which is blocked by *N*-acetyl-L-cysteine. Thus, ROS generation, as well as starvation, results in Beclin 1 S90

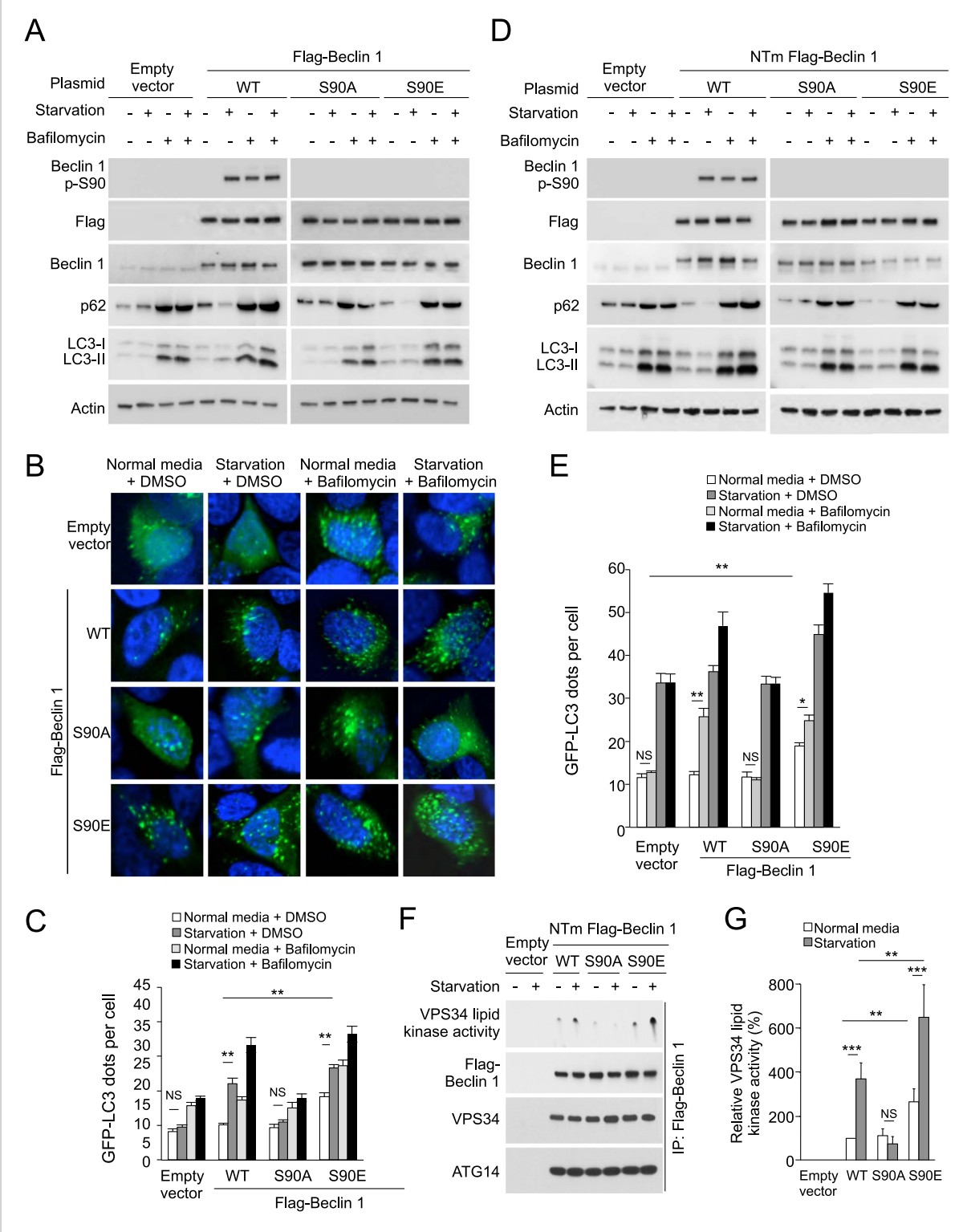

Figure 2. The Beclin 1 S90 phosphorylation site is required for autophagy induction in MCF7 and U2OS cells. (A) Western blot results of MCF7 cells transiently transfected with empty vector, and Flag epitope-tagged wild-type Beclin 1, Beclin 1 S90A, or Beclin 1 S90E. The cells were grown in normal medium (starvation−) or HBSS (starvation+) for 3 hr in the presence or absence of 100 nM bafilomycin A1. (B) Representative images of GFP-LC3 puncta (autophagosomes) in MCF7 cells transiently co-transfected with indicated Flag-Beclin 1 constructs and a plasmid expressing GFP-LC3 and grown in normal medium or in HBSS for 3 hr (starvation) in the presence or absence of 100 nM bafilomycin A1. (C) Quantification of GFP-LC3 puncta in MCF7 cells in conditions shown in (B). Bars are mean ± SEM of triplicate samples (≥50 cells analyzed per sample). Similar results were observed in three independent

Figure 2. continued on next page

*Figure 2. Continued*

experiments. ***p < 0.001, **p < 0.01, NS, not significant; one-way ANOVA. (**D**) Western blot detection of Beclin 1, p62 and LC3 in U2OS cells expressing doxycycline-inducible shRNA against *beclin 1* (*beclin 1* shRNA U2OS cells) following treatment with 1 μg/ml doxycycline for 4 days in cells transduced with retroviral constructs expressing indicated shRNA-resistant Flag-Beclin 1 (NTm, non-targetable mutant) plasmids. Cells were either grown in normal medium (starvation−) or in HBSS for 3 hr (starvation+) in the presence or absence of 100 nM bafilomycin A1. See *Figure 2—figure supplement 1* for comparison of Beclin 1, p62, and LC3 western blots in the presence and absence of doxycycline. (**E**) Quantification of GFP-LC3 puncta (autophagosomes) in *beclin 1* shRNA U2OS cells treated with 1 μg/ml doxycycline for 4 days and co-transfected with plasmids expressing GFP-LC3 and indicated shRNA-resistant Flag-Beclin 1 construct and grown in normal medium or in EBSS for 3 hr (starvation) in the presence or absence of 100 nM bafilomycin A1. Bars are mean ± SEM of triplicate samples (≥50 cells analyzed per sample). Similar results were observed in three independent experiments. **p < 0.01, *p < 0.05, NS, not significant; one-way ANOVA. (**F**) Beclin 1-associated VPS34 in vitro lipid kinase assay and amounts of VPS34 and ATG14 in anti-Beclin 1 immunoprecipitates of *beclin 1* shRNA U2OS cells following treatment with 1 μg/ml doxycycline for 4 days and transfection with indicated shRNA-resistant Flag-Beclin 1 (NTm, non-targetable mutant) plasmids. Cells were either grown in normal medium (starvation−) or in HBSS for 2 hr (starvation+). Dots shown in upper panel represent the amount of PI3P generated in an in vitro VPS34 lipid kinase assay using anti-Flag-Beclin 1 immunoprecipitates as input. (**G**) Densometric quantitation of VPS34 in vitro lipid kinase activity in anti-Beclin 1 immunoprecipitates in conditions described in (**F**). Results shown represent mean + SEM of values in three independent experiments. Similar results were observed in each independent experiment. Shown are the relative values of VPS34 lipid kinase activity compared to those observed in cells expressing WT Beclin 1 in normal media (defined as 100%). To control for input in Beclin 1 anti-immunoprecipitates, values used to calculate VPS34 lipid kinase activity were normalized for levels of Beclin 1 determined by densitometric quantification of Beclin 1 western blot bands in anti-Beclin 1 immunoprecipitates. ***p < 0.001, **p < 0.01, NS, not significant; one-way ANOVA. See also *Figure 2—figure supplement 1*, *Figure 1—figure supplement 2*, *Figure 2—figure supplement 3*.

The following figure supplements are available for figure 2:

**Figure supplement 1**. Doxycycline reduces Beclin 1 expression and starvation-induced autophagy in U2OS cells that express doxycycline-inducible *beclin 1* shRNA.

**Figure supplement 2**. Bafilomycin A1-induced Beclin 1 S90 phosphorylation is reversed by the ROS scavenger, *N*-acetyl-L-cysteine.

**Figure supplement 3**. Effects of Beclin 1 S90 phosphorylation o on subcellular localization of the Beclin 1/VPS34/ATG14 complex.

phosphorylation. We also observed Beclin 1 S90 phosphorylation in response to other stress stimuli (data not shown), but chose to focus on starvation, given the crucial physiological importance of starvation in autophagy induction.

To address the mechanism by which Beclin 1 S90 phosphorylation induces autophagy, we compared U2OS cells expressing wild-type Beclin 1, the non-phosphorylatable Beclin 1 S90A mutant, and the phosphomimetic Beclin 1 S90E mutant with respect to (1) Beclin 1-associated VPS34 lipid kinase activity; (2) Beclin 1 immunoprecipitation of ATG14 and VPS34, and (3) the membrane localization of Beclin 1, ATG14, and VPS34. Our results show that the amounts of VPS34 and ATG14 that co-immunoprecipitate with Beclin 1 are not altered by starvation or by mutation of the Beclin 1 S90 site (*Figure 2F*). In addition, no apparent differences were observed in the localization of Beclin 1, ATG14, or VPS34 to membrane fractions (e.g., mitochondrial-enriched, microsomal) in cells expressing wild-type Beclin 1, Beclin 1 S90A, or Beclin 1 S90E (*Figure 2—figure supplement 3*). However, despite similar levels of Beclin 1/ATG14 and Beclin 1/VPS34 binding, marked differences were observed in the VPS34 lipid kinase activity associated with wild-type and mutant forms of Beclin 1 (*Figure 2F–G*). In cells expressing the phosphorylation-defective Beclin 1 S90A mutant, no increase was observed in Beclin 1-associated VPS34 lipid kinase activity in response to starvation. In cells expressing the Beclin 1 S90E phosphomimetic mutant, Beclin 1-associated VPS34 lipid kinase activity was higher in baseline and in starvation conditions than in cells expressing wild-type Beclin 1. Thus, Beclin 1 S90 phoshphorylation increases the lipid kinase activity of the Beclin 1/VPS34 complex.

## The Beclin 1 S90 phosphorylation site is required for tumor suppressor function

Beclin 1 is a tumor suppressor protein that inhibits the growth of MCF7 human breast carcinoma cells in immunodeficient mice (*Liang et al., 1999*, *2001*; *Furuya et al., 2005*). We investigated whether the Beclin 1 S90 phosphorylation site required for starvation-induced autophagy is also required for the tumor suppressor activity of Beclin 1. MCF7 cells were transduced with retroviruses that express Flag epitope-tagged wild-type Beclin 1 or Beclin 1 S90A; injected into *nu/nu* mice implanted with

slow release estrogen tablets; and monitored for their rate of tumor growth. Levels of Beclin 1 expression were comparable in MCF7 cells transduced with both Beclin 1-expressing viruses (*Figure 3A*). Even in normal growth conditions, Beclin 1 S90 phosphorylation could be detected in MCF7 cells expressing wild-type Beclin 1, presumably due to high levels of expression of the protein using a retroviral vector; such phosphorylation was absent in MCF7 cells transduced with a retrovirus expressing mutant Beclin 1 S90A.

Striking differences in the rate of MCF7 xenograft growth in nude mice were observed in cells expressing wild-type Beclin 1 as compared to Beclin 1 S90A; MCF7 cells expressing Beclin 1 S90A grew as rapidly as MCF7 control cells, whereas a marked reduction in tumor growth was observed in MCF7 cells expressing wild-type Beclin 1 (*Figure 3B*). In xenografts expressing wild-type Beclin 1, but not Beclin 1 S90A, there was a significant reduction in tumor p62 staining as compared with empty vector controls (*Figure 3C,D*), suggesting increased levels of autophagy in the tumors. In parallel with decreased growth and decreased p62 staining, xenografts expressing wild-type Beclin 1, but not Beclin 1 S90A, had decreased rates of cell proliferation (*Figure 3C,E*), as measured by the percentage of cells with Ki67 staining. Xenografts expressing wild-type Beclin 1 also had decreased cell death (*Figure 3C,F*), as measured by the percentage of TUNEL-positive cells. Together, these data confirm previous reports demonstrating that Beclin 1 simultaneously increases cell survival and decreases cell proliferation, resulting in a net decrease in tumor cell growth (*Wei et al., 2013*), and provide new evidence that the S90 phosphorylation site is essential for the tumor suppressor function of Beclin 1.

## Beclin 1 S90 is phosphorylated by MAPKAPK2 (MK2) and MAPKAPK3 (MK3)

Next, we sought to identify the upstream kinase(s) that mediate starvation-induced phosphorylation of Beclin 1 S90. Therefore, we performed an in vitro kinase screen using a 15 amino acid peptide corresponding to Beclin 1 amino acid residues 83–97 as a substrate. Among a pool of 190 serine/threonine kinases, MAPKAPK3, also known as and herein referred to as MK3, was the only kinase with significant activity against the Beclin 1 peptide substrate (*Figure 4—figure supplement 1*, *Supplementary file 1*). Since there are other serine and threonine residues in the Beclin 1 83–97 peptide, we repeated the in vitro kinase assay with MK3 using a Beclin 1 peptide spanning from amino acid residues 83–103 with either a wild-type sequence or an alanine substitution mutation at amino acid residue 90. This mutation completely abrogated MK3-mediated phosphorylation of the Beclin 1 peptide, indicating that serine 90 is essential for it to serve as a substrate for MK3 (*Figure 4A*).

MK3 is highly homologous to MK2, another MAP kinase-activated protein kinase, and shares ~75% sequence identity and displays functional redundancy (*Gaestel, 2006*). We found that MK2, but not MK5—another MAP kinase-activated protein kinase that is less structurally related to MK3 (*Cargnello and Roux, 2011*)—also phosphorylated the Beclin 1 83–97 peptide in vitro (*Figure 4B*). Furthermore, both MK2 and MK3 demonstrated in vitro kinase activity, resulting in Beclin 1 S90 phosphorylation, using Flag-Beclin 1 purified from HEK293T cells as a substrate (*Figure 4C*). To investigate whether MK2 phosphorylates Beclin 1 S90 in cultured cells, we expressed either dominant-negative MK2 (MK2 K/R) (containing an arginine substitution at a lysine position at amino acid residue 76 that is conserved between MK2 and MK3), (*Winzen et al., 1999*) or constitutively active MK2 (MK2 T205E/T317E; herein referred to as MK2 EE) (*Engel et al., 1995*) in HeLa cells. Dominant-negative MK2 blocked starvation-induced Beclin 1 S90 phosphorylation, whereas constitutively active MK2 increased Beclin 1 S90 phosphorylation during nutrient-rich conditions (*Figure 5A*). By measuring phosphorylation of the downstream MK2/MK3 substrate, p-HSP27, we confirmed that starvation activated MK2/MK3 and that this was blocked by dominant negative MK2 expression. Taken together, these data indicate that MK2 and MK3 directly phosphorylate Beclin 1 S90 in vitro, and that MK2-like kinase activity also mediates starvation-induced Beclin 1 S90 phosphorylation in cultured cells.

## MK2 positively regulates starvation-induced autophagy

To study the effects of MK2 and MK3 on autophagy regulation, we first evaluated autophagy in HeLa cells transfected with dominant negative MK2 (which blocks the kinase activity of both MK2 and MK3) or constitutively active MK2. As measured by LC3-II conversion, p62 degradation, and numbers of GFP-LC3 puncta in the presence and absence of bafilomycin A1, dominant-negative MK2 decreased starvation-induced autophagy (*Figure 5A,B*). Conversely, constitutively active MK2 increased basal

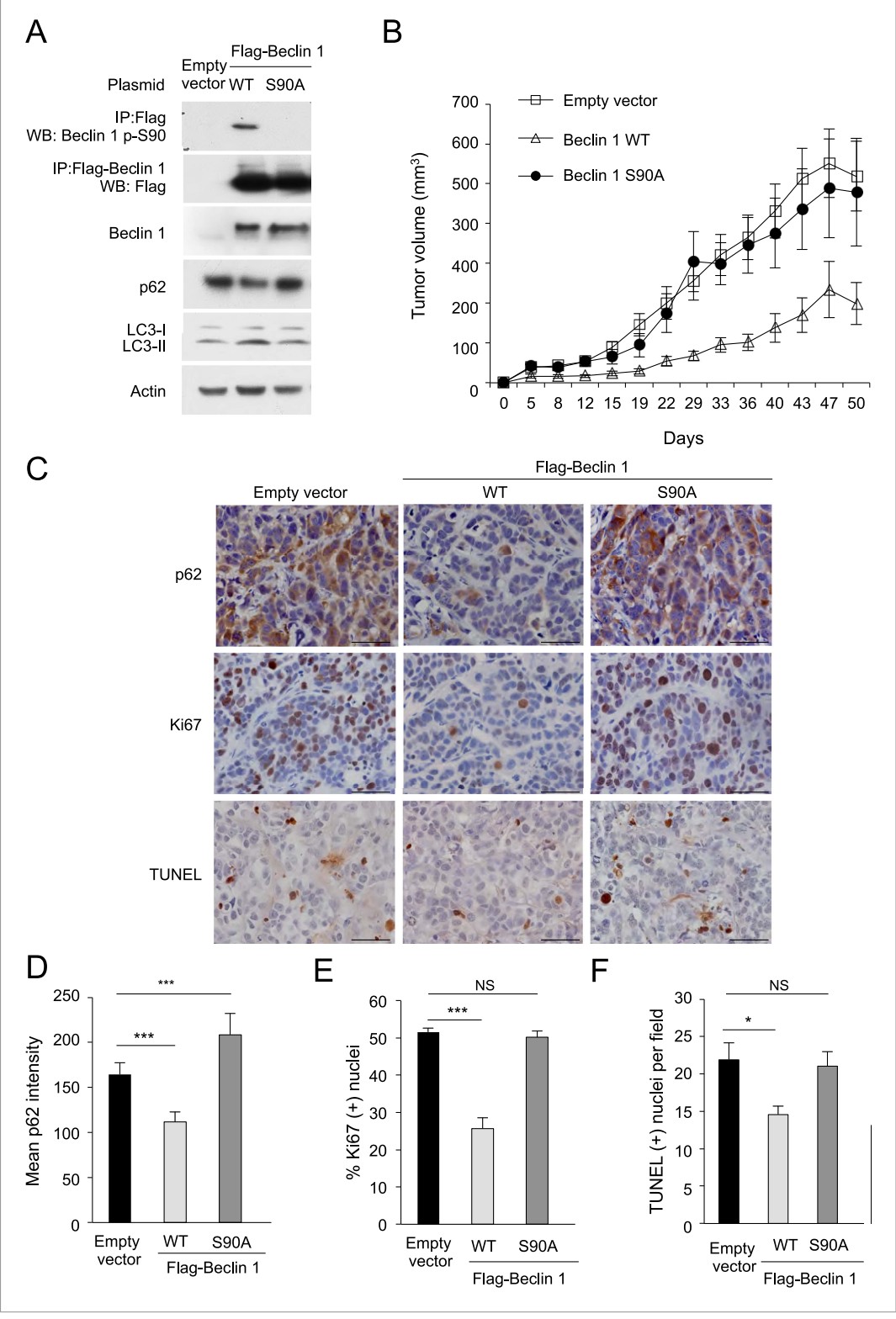

**Figure 3**. The Beclin 1 S90 phosphorylation site is required for tumor suppression function in MCF7 cells. (**A**) Western blot detection of Beclin 1 p-S90, Flag-Beclin 1, p62, and LC3 in MCF7 cells stably transduced with retroviruses expressing indicated Beclin 1 construct. (**B**) Xenograft growth of cells in (**A**) in *nu/nu* mice. Tumor volumes shown represent mean ± SEM for at least 11 mice per group. Differences in tumor growth among the different groups were analyzed using a linear mixed effect model; *P* = NS for Beclin 1 S90A vs empty vector; *Figure 3. continued on next page*

*Figure 3. Continued*

p < 0.001 for Beclin 1 WT vs empty vector. (**C**) Representative images of p62 staining, Ki67 staining, and TUNEL-labeling of indicated MCF7 xenograft tumor genotype. (**D–F**) Quantification of relative reciprocal p62 intensity per high-power field (**D**), percentage of Ki67-positive nuclei per high-power field (**E**) and number of TUNEL-positive nuclei per high power field (**F**). At least 10 randomly selected fields per xenograft section were analyzed by an observer blinded to genotype. Bars are mean ± SEM for 5–8 xenografts per MCF7 xenograft tumor genotype. \*\*\*p < 0.001, \*\*p < 0.01, \*p < 0.05, NS, not significant; one-way ANOVA with Dunnett's test.

autophagy in HeLa cells (*Figure 5A,B*). Thus, in parallel with regulation of starvation-induced Beclin 1 S90 phosphorylation (*Figure 5A*), MK2 regulates starvation-induced autophagy.

The best-characterized kinase upstream of MK2/MK3 is p38 MAPK (*Freshney et al., 1994*; *Rouse et al., 1994*). Therefore, we asked whether dominant negative p38α could block starvation-induced MK2 activation, Beclin 1 S90 phosphorylation, and autophagy. In HeLa cells transfected with a plasmid expressing dominant negative p38α, we failed to observe starvation-induced MK2 activation (i.e., an MK2 band shift or phosphorylation of its substrate HSP27), Beclin 1 S90 phosphorylation, or autophagy (*Figure 5—figure supplement 1A*). This block in Beclin 1 S90 phosphorylation in cells expressing dominant negative p38α was reversed by overexpression of constitutively active MK2 (*Figure 5—figure supplement 1B*), providing additional proof that MK2 functions downstream of p38α to mediate Beclin 1 S90 phosphorylation in response to amino acid starvation.

To determine whether endogenous MK2/MK3 function in the regulation of Beclin 1 S90 phosphorylation and autophagy, we evaluated *Mapkapk2*[−/−]/*Mapkapk3*[−/−] (herein referred to as *MK2*[−/−]/*MK3*[−/−]) MEFs retrovirally transduced with empty vector compared to *MK2*[−/−]/*MK3*[−/−] MEFs retrovirally transduced with MK2 (*Ronkina et al., 2011*). In *MK2*[−/−]/*MK3*[−/−] MEFs, there was a marked decrease in starvation-induced Beclin 1 S90 phosphorylation which was rescued by MK2 expression (*Figure 5C*). Low levels of Beclin 1 S90 phosphorylation were still detected in *MK2*[−/−]/*MK3*[−/−] MEFs following starvation, suggesting either that other members of the MAPKAPK family may partially compensate for the developmental loss of both *MK2* and *MK3* and/or that other unidentified kinase families may also play a minor role in the regulation of phosphorylation at this site of Beclin 1. Nonetheless, loss of *MK2* and *MK3* was sufficient to decrease both basal and starvation-induced autophagy as assessed by p62 degradation, LC3-II conversion, and quantification of GFP-LC3 puncta in *MK2*[−/−]/*MK3*[−/−] MEFs as compared to *MK2*[−/−]/*MK3*[−/−] MEFs that stably express MK2 (*Figure 5C,D*). The increase in GFP-LC3 puncta with MK2 reconstitution was not due to a block in autophagic maturation, as numbers further increased upon treatment with the lysosomal inhibitor, bafilomycin A1. Also, we note that the decreased levels of total LC3 (with an increased ratio of LC3-II to LC3-I) in MK2 reconstituted cells during starvation conditions is consistent with a marked increase in autophagic flux, as LC3 (similar to p62) is degraded by the autophagy pathway. Thus, these results in *MK2*[−/−]/*MK3*[−/−] MEFs demonstrate that endogenous MK2 and MK3 function in the positive regulation of Beclin 1 S90 phosphorylation and autophagy.

## MK2 positively regulates starvation-induced autophagy through a mechanism involving the Beclin 1 S90 phosphorylation site

To examine whether MK2/3 positively regulate autophagy through a mechanism that involves Beclin 1 S90 phosphorylation, we examined the effects of retroviral gene transfer of wild-type Beclin 1, Beclin 1 S90A (a non-phosphorylatable mutant), or Beclin 1 S90E (a predicted phosphomimetic mutant) on autophagy in *MK2*[−/−]/*MK3*[−/−] MEFs. Whereas expression of Beclin 1 S90A did not increase levels of basal or starvation-induced autophagy (and expression of wild-type Beclin 1 had only minimal effects), Beclin 1 S90E expression was sufficient to markedly increase basal autophagy in cells lacking both MK2 and MK3 (*Figure 6A,B*). The magnitude of this increase was almost comparable to that observed with enforced expression of MK2. The observation that a Beclin 1 S90 phosphomimetic mutant can substitute for MK2/3 in MK2/3-deficient cells in regulating levels of autophagy is consistent with a model in which MK2/3 regulates autophagy through the phosphorylation of Beclin 1 S90.

To more directly test the hypothesis that MK2/3 regulate autophagy through the Beclin 1 S90 phosphorylation site, we investigated whether active MK2 can induce autophagy in cells that lack

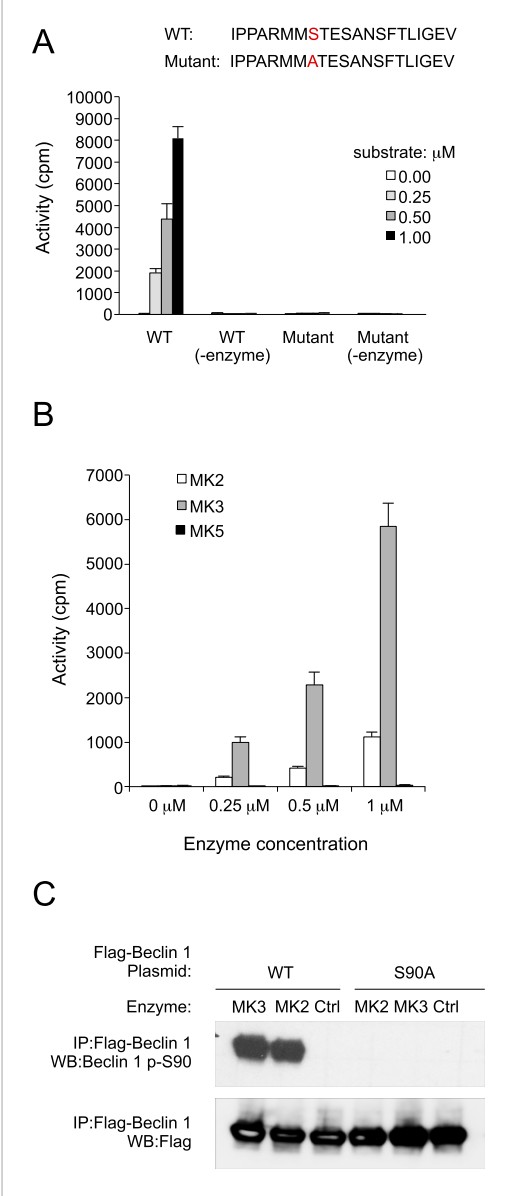

**Figure 4**. MK2 and MK3 mediate phosphorylation of Beclin 1 S90. (**A**) In vitro kinase activity of MK3 using indicated Beclin 1 peptides as a substrate. (**B**) In vitro kinase activity of MK2, MK3, and MK5 using the wild-type (WT) Beclin 1 peptide shown in A as a substrate. (**C**) In vitro kinase activity of MK2 and MK3 using Flag epitope-tagged wild-type Beclin 1 or Beclin 1 S90 purified from HEK293T cells as a substrate. See also *Figure 4—figure supplement 1* and *Supplementary file 1*.

The following figure supplement is available for figure 4:

**Figure supplement 1**. Results of in vitro kinase screen using the Beclin 1 83–97 peptide as a substrate.

detectable endogenous Beclin 1 and only express an shRNA-resistant mutant Beclin 1 S90A construct (*Figure 6C*). U2OS cells with doxycycline treatment-induced knockdown of Beclin 1 fail to undergo increased autophagy (as assessed by p62 degradation and LC3-II conversion) in response to starvation, active MK2 expression, or both starvation and active MK2 expression. In cells that express an shRNA-resistant wild-type Beclin 1, there is an increase in autophagy with starvation and active MK2 expression, and a further increase in autophagy with combined starvation and active MK2 expression. In contrast, in cells that express an shRNA resistant Beclin 1 S90A mutant construct, no increase in autophagy is observed in response to any of these treatment conditions. Thus, MK2-induced upregulation of autophagy requires the Beclin 1 S90 phosphorylation site.

Conversely, we asked whether expression of a Beclin 1 S90E phosphomimetic mutant in cells lacking detectable endogenous Beclin 1 could bypass the inhibitory effects of dominant negative MK2 (*Figure 6D*). In doxycline-treated *beclin 1* shRNA U2OS cells that express shRNA-resistant wild-type Beclin 1, dominant negative MK2 nearly completely suppressed starvation-induced autophagy, as assessed by p62 degradation and LC3-II conversion. However, in cells expressing shRNA-resistant Beclin 1 S90E, starvation induced a decrease in p62 levels and conversion of LC3-I to LC3-II even in the presence of dominant negative MK2 expression. This suggests that Beclin 1 S90 phosphorylation may be sufficient to partially bypass the inhibitory effects of MK2/3 inhibition on starvation-induced autophagy.

Taken together, these data in $MK2^{-/-}/MK3^{-/-}$ MEFs and in doxycycline-inducible *beclin 1* shRNA U2OS cells (reconstituted with wild-type Beclin 1, the non-phosphorylatable Beclin 1 S90A mutant or the phosphomimetic Beclin 1 S90E mutant) strongly suggest that MK2/3 regulate autophagy through a mechanism involving Beclin 1 S90 phosphorylation.

## BCL2 negatively regulates MK2-Mediated Beclin 1 serine 90 phosphorylation

BCL2 inhibits autophagy through a direct interaction with the BH3 domain of Beclin 1, which consists of amino acid residues 105–128 (*Oberstein et al., 2007*). Given the proximity of the BH3 domain to Beclin 1 S90, we postulated that BCL2 binding to Beclin 1 might affect MK2-dependent Beclin 1 phosphorylation. In addition, the possibility that BCL2 binding to Beclin 1 (a process disrupted by JNK1-mediated multisite phosphorylation of BCL2 during amino acid starvation [*Wei et al., 2008*]) might block

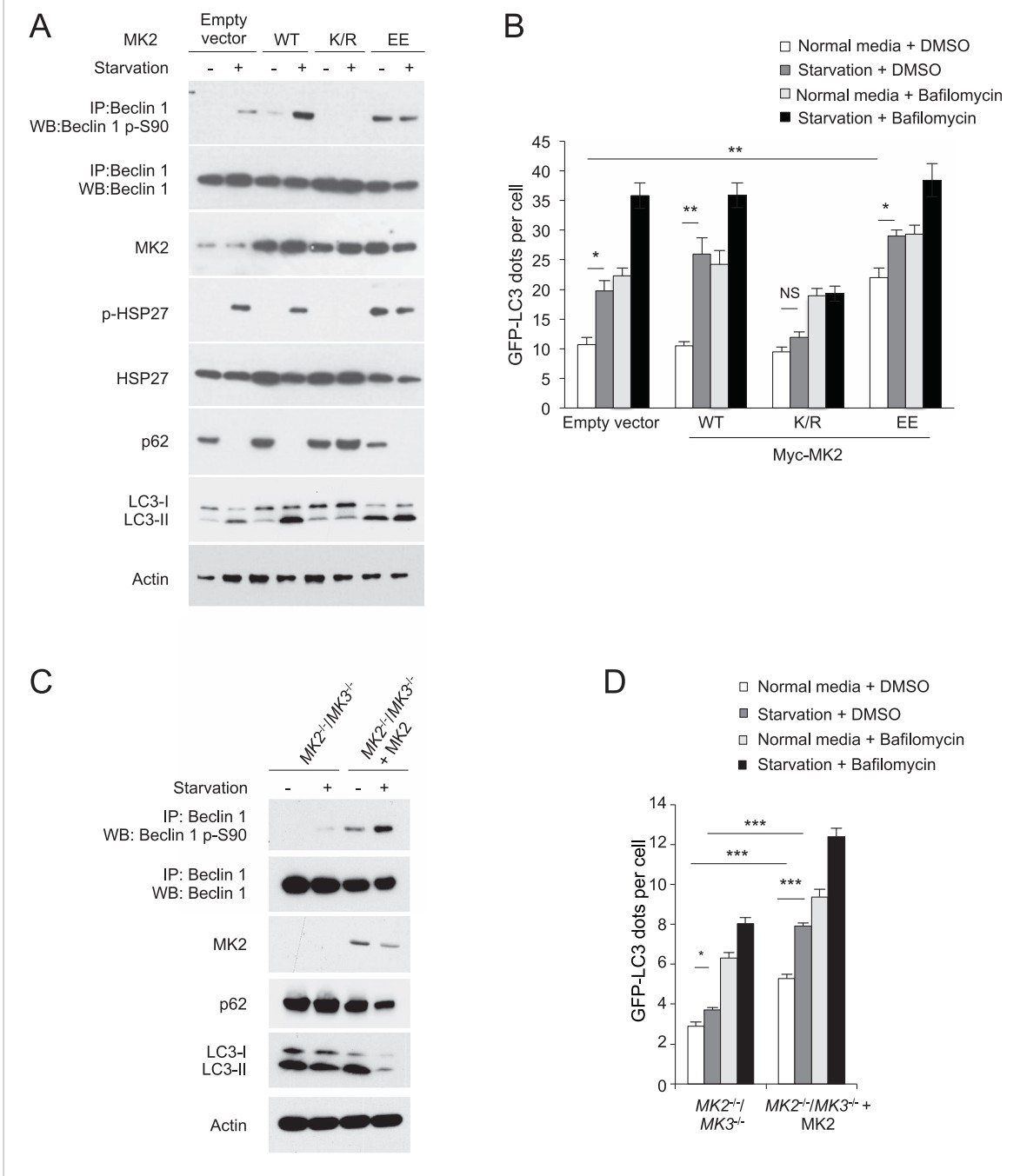

**Figure 5**. MK2 positively regulates autophagy. (**A**) Effects of wild-type, dominant negative (K/R) and constitutively active (EE) MK2 on endogenous Beclin 1 S90 phosphorylation, MK2/MK3 activation (levels of p-HSP27) and autophagy (levels of p62 degradation and LC3-II conversion) in HeLa cells grown in normal medium (starvation−) or HBSS for 2 hr (starvation+). Actin is shown as a loading control. (**B**) Effects of wild-type, dominative negative (K/R) and constitutively active (EE) MK2 on autophagy (GFP-LC3 puncta numbers in the presence or absence of 100 nM bafilomycin A1) in HeLa cells grown in normal medium (starvation−) or HBSS for 3 hr (starvation+). Bars are mean $\pm$ SEM of triplicate samples ($\geq$50 cells analyzed per sample). Similar results were observed in three independent experiments. **$p < 0.01$, *$p < 0.05$, NS, not significant; one-way ANOVA for indicated comparisons. (**C**) Beclin 1 S90 phosphorylation and autophagy (levels of p62 degradation and LC3-II conversion) in $MK2^{-/-}/MK3^{-/-}$ MEFs and $MK2^{-/-}/MK3^{-/-}$ MEFs stably transformed with wild-type MK2. Cells were grown in normal medium (starvation−) or HBSS for 2 hr (starvation+). (**D**) Quantitation of GFP-LC3 puncta (autophagosomes) in $MK2^{-/-}/MK3^{-/-}$ MEFs and $MK2^{-/-}/MK3^{-/-}$ MEFs stably transformed with wild-type MK2 during growth in normal media or HBSS (starvation) for 3 hr in the presence or absence of 100 nM bafilomycin A1. Bars are mean $\pm$ SEM of triplicate samples ($\geq$50 cells analyzed per sample). Similar results were observed in three independent experiments. ***$p < 0.001$, NS, not significant; one-way ANOVA. $p < 0.001$ for the magnitude of change between normal and starvation conditions in $MK2^{-/-}/MK3^{-/-}$ MEFs vs that in

*Figure 5. continued on next page*

*Figure 5. Continued*

in *MK2⁻/⁻/MK3⁻/⁻* + MK2 MEFs; two-way ANOVA. See also *Figure 5—figure supplement 1*.

The following figure supplement is available for figure 5:

**Figure supplement 1**. Dominant p38α inhibits starvation-induced MK2 activation and Beclin 1 S90 phosphorylation.

MK2-dependent phosphorylation was suggested by two observations in this study, including (1) dominant-negative JNK1 blocked starvation-induced Beclin 1 S90 phosphorylation and autophagy but not starvation-induced MK2 activation (*Figure 5—figure supplement 1*); and (2) the amount of active MK2-dependent Beclin 1 S90 phosphorylation was greater in starvation conditions than in normal nutrient conditions (*Figure 6C*).

To directly test the hypothesis that BCL2 binding to Beclin 1 blocks MK2-mediated Beclin 1 S90 phosphorylation, we performed an in vitro kinase assay to assess the effects of MK2 on Beclin 1 S90 phosphorylation in the presence of recombinant GST-BCL2. We found that with increasing amounts of GST-BCL2, there was decreased in vitro MK2-mediated phosphorylation of Beclin 1 S90 (*Figure 7A*). These data provide direct evidence that BCL2 inhibits MK2-dependent Beclin 1 S90 phosphorylation.

To better understand how binding of BCL2 family proteins to Beclin 1 may inhibit MK2-mediated phosphorylation, we constructed a structure-based model (*Figure 7B*) to determine whether both BCL2 and MK2 could bind simultaneously to Beclin 1. The structure of the BCL2-related anti-apoptotic protein, BCL2L1 (which also inhibits autophagy through its interaction with Beclin 1), bound to the helical BH3 domain of Beclin 1 has been solved (*Oberstein et al., 2007*) (PDV ID: 2P1L), as has the structure of a region of MK2 bound to its auto-inhibitory peptide (*Meng et al., 2002*) (PDB ID: 1KWP). Based on sequence analysis suggesting that Beclin 1 residues 80–104 that precede the BH3 domain are helical when bound to a suitable partner (*Mei et al., 2014*), a model for this region was built by 'in silico' mutagenesis of the MK2-bound auto-inhibitory peptide to correspond to Beclin 1 residues 80–95, such that S90 was positioned optimally for phosphorylation. This helix was then extended to include Beclin 1 residues 96–131 and BCL2L1 positioned by superposition of the bound Beclin 1 BH3 domain (residues 105–128) on the modeled Beclin 1 extended helix. This model suggests that steric hindrance may prevent simultaneous binding of BCL2L1 (or BCL2) and MK2 to Beclin 1, thereby inhibiting MK2-dependent Beclin 1 S90 phosphorylation. Of note, residues 216–236 and the N- and C-terminal regions of MK2 and BCL2L1 residues 26–81, were not present in the structures used to build this model, but in the full-length proteins these residues likely result in additional steric conflicts (*Figure 7B*). Furthermore, the BCL2L1 C-terminal membrane-anchoring helix that was removed in the construct crystallized may provide cellular localization constraints that prevent MK2-mediated phosphorylation of Beclin 1 S90.

Next, we assessed whether BCL2 binding to Beclin 1 in cultured cells regulates Beclin 1 S90 phosphorylation (*Figure 7C,D*). In HeLa cells stably transfected with wild-type BCL2, the starvation-induced increase in Beclin 1 S90 phosphorylation was delayed and blunted compared to that observed in HeLa control cells; however, by 2 hr after starvation when there was complete disruption of BCL2 and Beclin 1 binding (as measured by co-immunoprecipitation), an appreciable increase in Beclin 1 S90 phosphorylation was observed (*Figure 7C*). In contrast, in HeLa cells stably transfected with the mutant BCL2 T69A/S70A/S87A (BCL2 AAA), which lacks the JNK1 phosphorylation sites and fails to dissociate with Beclin 1 during starvation (*Wei et al., 2008*), there was no increase in Beclin 1 S90 phosphorylation in response to starvation. The levels of starvation-induced autophagy, as assessed by p62 degradation and LC3-II conversion paralleled the levels of Beclin 1 S90 phosphorylation. Starvation-induced autophagy was delayed and attenuated in HeLa/BCL2 cells compared to HeLa/control cells, and completely blocked in HeLa/BCL2 AAA cells. These effects of BCL2 on starvation-induced Beclin 1 S90 phosphorylation and autophagy were not due to a blockade of MK2/MK3 activation, as levels of phospho-HSP27 increased similarly during starvation in HeLa/control, HeLa/BCL2 and HeLa/BCL2 AAA cells. Furthermore, even in the absence of starvation, active MK2 was able to increase Beclin 1 S90 phosphorylation and induce autophagy in HeLa/control cells but not in HeLa/BCL2 or HeLa/BCL2 AAA cells (*Figure 7D*). Thus, BCL2 blocks both starvation- and active MK2-mediated upregulation of Beclin 1 S90 phosphorylation and autophagy.

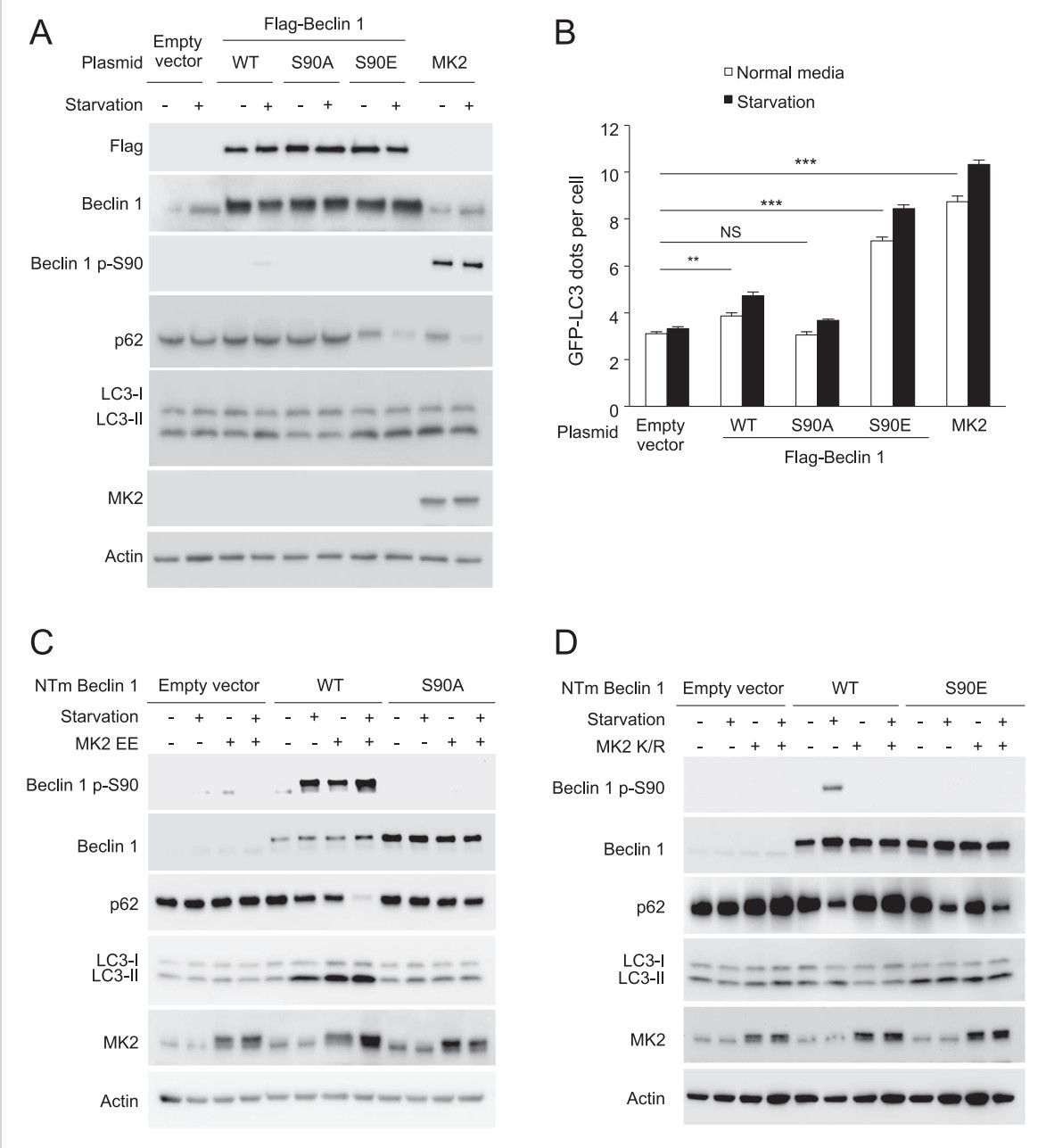

**Figure 6**. MK2 positively regulates autophagy through the Beclin 1 S90 phosphorylation site. (**A**) Western blot detection of Beclin 1 S90 phosphorylation and autophagy (p62 degradation and LC3-II conversion) in $MK2^{-/-}/MK3^{-/-}$ MEFs retrovirally transduced with indicated expression constructs and grown in normal media or HBSS (starvation) for 4 hr. (**B**) Quantification of GFP-LC3 puncta (autophagosomes) in $MK2^{-/-}/MK3^{-/-}$ MEFs retrovirally transduced with indicated expression constructs and grown in normal media or HBSS (starvation) for 4 hr. Bars are mean $\pm$ SEM of triplicate samples ($\geq$50 cells analyzed per sample). Similar results were obtained in two independent experiments. ***$p < 0.01$, NS, not-significant; one-way ANOVA with Dunnett's test. (**C**) Effects of constitutively active MK2 (MK2 EE) on Beclin 1 S90 phosphorylation and autophagy (as measured by p62 and LC3 western blot analysis) in U2OS doxycycline-inducible *beclin 1* shRNA knockdown cells expressing either shRNA-resistant wild-type Flag-Beclin 1 or mutant Flag-Beclin 1 S90A. Cells were treated with 1 µg/ml doxycycline for 4 days prior to western blot analyses with indicated antibodies and transfected with indicated shRNA-resistant plasmids after 2 days of doxycycline treatment. Cells were grown in normal media or HBSS (starvation) for 2 hr. NTm, non-targeting mutant. (**D**) Effects of dominant-negative MK2 (MK2 K/R) on Beclin 1 S90 phosphorylation and autophagy (as measured by p62 and LC3 western blot analysis) in U2OS doxycycline-inducible *beclin 1* shRNA knockdown cells expressing either shRNA-resistant wild-type Flag-Beclin 1 or mutant Flag-Beclin 1 S90A. Cells were treated with 1 µg/ml doxycycline for 4 days prior to western blot analyses with indicated antibodies, and transfected with indicated shRNA-resistant plasmids after 2 days of doxycycline treatment. Cells were grown in normal media or HBSS (starvation) for 2 hr. NTm, non-targeting mutant.

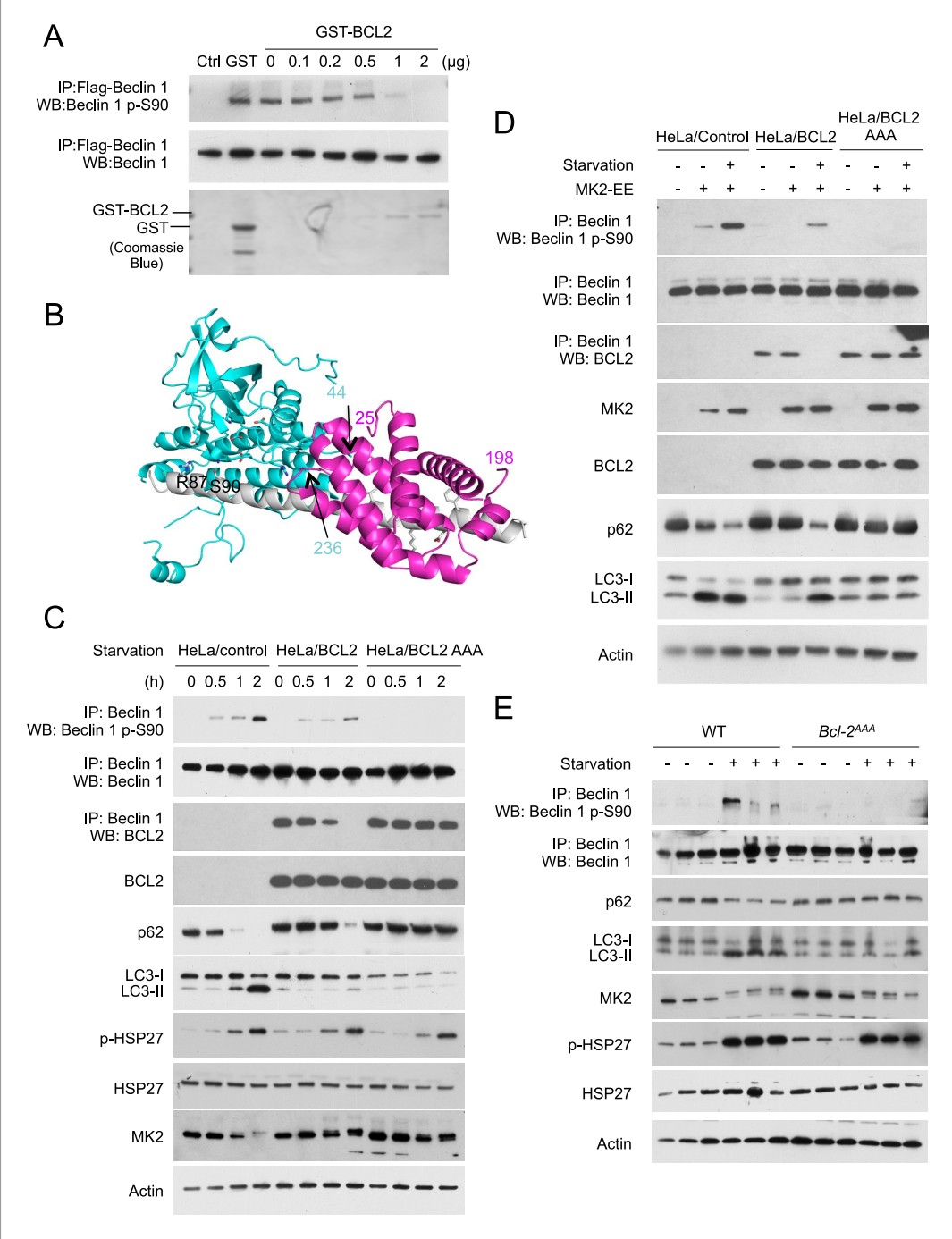

**Figure 7**. BCL2 inhibits MK2-dependent Beclin 1 S90 phosphorylation. (**A**) In vitro kinase activity of MK2 using Flag-Beclin 1 purified from HEK293 cells as a substrate in the presence of the indicated amount of GST-BCL2. (**B**) Structural model showing that steric hindrance may prevent simultaneous binding of BCL2/BCL2L1 and MK2 to Beclin 1. MK2 (residues 44–216, 236–284 and 288–345), BCL2L1 (residues 2–25 and 83–198) and Beclin 1 (residues 79–181) are shown in cyan, pink and grey ribbon, respectively. MK2 residues important for catalysis and Beclin 1 residues important for binding to BCL2L1, or for phosphorylation (S90 and R87) are shown in stick, with atoms colored by element type: oxygen, red; nitrogen, blue; sulfur, green; and carbon, cyan or grey for MK2 or Beclin 1 respectively. Numbers indicate the last residue preceding or following regions missing from this model, that might cause further steric conflicts. (**C**) Detection of Beclin 1 S90 phosphorylation and autophagy levels (as measured by p62 and LC3 immunoblots) in HeLa cells stably transfected with empty vector (HeLa/control), wild-type BCL2 (HeLa/BCL2), or a BCL2 T69A/S70A/S87A mutant (HeLa/BCL2 AAA) and subjected to starvation in HBSS for the indicated

*Figure 7. continued on next page*

*Figure 7. Continued*

time period. p-Hsp27 protein levels are shown as an indicator of MK2 activation. (**D**) Detection of Beclin 1 S90 phosphorylation and autophagy levels (as measured by p62 and LC3 immunoblots) in HeLa/Control, HeLa/BCL2, and HeLa/BCL2 AAA cells transiently transfected with a control empty vector or constitutively active MK2 (MK2 EE) and grown in normal medium (starvation−) or HBSS for 2 hr (starvation+). (**E**) Starvation-induced Beclin 1 S90 phosphorylation and autophagy induction in vastus lateralis muscle of *Bcl2*^AAA mice and control wild-type littermates. Mice were subjected to starvation for 48 hr prior to tissue collection and western blot analysis of muscle lysates with indicated antibodies. Each lane represents a muscle sample from an independent mouse.

To confirm whether BCL2 regulates starvation-induced Beclin 1 S90 phosphorylation in vivo, we examined the levels of Beclin 1 S90 phosphorylation in *Bcl2*^AAA mice that contain a T69A/S70A/S84A knock-in mutation in Bcl2 (S84 is homologous to human S87) (*Figure 7E*). Previously, we showed that this mutation blocks starvation-induced disruption of Bcl2/Beclin 1 binding and starvation-induced autophagy, including in cardiac and skeletal muscle of *Bcl2*^AAA mice (*He et al., 2012*). Following a 48 hr starvation period, we found the skeletal muscle of wild-type mice had activation of MK2 (as evidenced by a band shift in MK2 and phosphorylation of the substrate, HSP27), increased autophagy (as detected by p62 degradation and LC3-II conversion), and increased Beclin 1 S90 phosphorylation. In contrast, in the muscles of *Bcl2*^AAA mice, activation of MK2 appeared similar to that observed in wild-type muscles, but this was not accompanied by an increase in Beclin 1 S90 phosphorylation or in levels of autophagy. Thus, the constitutive binding of Bcl2 AAA to Beclin 1 during starvation in vivo is sufficient to block the effects of activated MK2 on Beclin 1 S90 phosphorylation. Based on our data above in cultured cells, we propose that this block in MK2-regulated Beclin 1 S90 phosphorylation may contribute to the mechanism by which Bcl2 inhibits autophagy in vivo.

## Discussion

### MK2/MK3 are crucial kinases for starvation-induced autophagy

Autophagy is a fundamental cellular survival response during nutrient starvation, permitting cells to maintain nutrient and energy homeostasis when external food supply is limited (*Levine and Klionsky, 2004*). The negative regulation of this process by mTORC1 has been extensively studied, but less is known about stress-activated signals that turn on autophagy during starvation (*Kroemer et al., 2010*). While AMPK activates components of the autophagy machinery during glucose starvation (phosphorylating ULK1, Beclin 1, and VPS34), it does not appear to regulate the Beclin 1/VPS34 complex in response to complete (nitrogen and glucose) starvation (*Kim et al., 2013*). The signaling molecules that activate Beclin 1/VPS34 to rapidly initiate autophagosome formation during complete starvation have been heretofore unknown.

Using unbiased approaches, we identified Beclin 1 S90 as a key phosphorylation site in starvation-induced autophagy and identified two stress-activated protein kinase signaling pathway family members, the p38 MAPK-activated protein kinases MK2 and MK3, as crucial kinases that mediate Beclin 1 S90 phosphorylation. A previous study showed that Beclin 1 S90 is phosphorylated during starvation and may function in autophagy (based on a ubiquitin-p62 degradation assay) (*Fogel et al., 2013*), but did not identify the kinase(s) responsible for this phosphorylation. In the present study, we found that alanine substitution of Beclin 1 serine 90 is sufficient to block detection of starvation-induced Beclin 1 phosphorylation, and sufficient to block Beclin 1-mediated rescue of starvation-induced autophagy in multiple different *beclin 1*-deficient cells using multiple different autophagic flux assays. The structurally and functionally-related p38 MAPK-activated protein kinase family members, MK2 and MK3 (*Cargnello and Roux, 2011*), but not the more distant family member, MK5, are strong inducers of Beclin 1 S90 phosphorylation. MK2 and MK3 phosphorylate a Beclin 1 peptide spanning S90 in vitro; MK2 and MK3 phosphorylate Beclin 1 immunoprecipitated from cells in vitro; dominant negative MK2 blocks starvation-induced Beclin 1 S90 phosphorylation and constitutively active MK2 induces Beclin 1 S90 starvation during basal conditions; and MEFs from *MK2/MK3* knockout mice are deficient in starvation-induced Beclin 1 S90 phosphorylation. Moreover, MK2 positively regulates autophagy through a mechanism that requires the Beclin 1 S90 phosphorylation site.

These findings describe a previously unidentified direct link between specific members of the MAPK signaling pathway and the positive regulation of a core component of the autophagy

machine during starvation. MK2 was originally discovered as an ERK1/2 activated protein kinase that phosphorylates HSP25 and HSP27 (*Stokoe et al., 1992*), and was later found to be stimulated by p38 in response to stress stimuli (*Freshney et al., 1994*; *Rouse et al., 1994*). MK3 was discovered independently through a yeast two-hybrid screen for p38-interacting proteins (*McLaughlin et al., 1996*) and through an analysis of genes commonly deleted in small-cell lung cancer (*Sithanandam et al., 1996*). The substrate spectrum of MK2 and MK3 are virtually indistinguishable, and both enzymes modulate proteins involved in cytokine production, endocytosis, actin remodeling, cell migration, cell cycle control, chromatin remodeling, and transcriptional regulation (*Cargnello and Roux, 2011*). The role of MK2 and MK3 in LPS-induced inflammatory signaling is well-established in vivo; *MK2* knockout mice show increased resistance to endotoxic shock and increased susceptibility to infections (*Kotlyarov et al., 1999*; *Lehner et al., 2002*) and this phenotype is exacerbated by simultaneous deletion of *MK3* (*Ronkina et al., 2007*). To our knowledge, MK2 and MK3 have not been previously linked to starvation-induced stress responses or to phosphorylation of specific substrates involved in starvation-induced stress response pathways such as autophagy. The newly described role of these kinases in regulating starvation-induced autophagy raises the possibility that this arm of the MAPKAPK family (sometimes referred to as the 'p38 module') may have evolved to integrate autophagy induction with diverse other cellular functions required for cellular adaption to stress.

Our data also suggest that MK2/MK3 are activated during amino acid starvation by the well-characterized upstream kinase, p38α. Several reports have shown that p38 functions in the positive regulation of autophagy in response to other stress stimuli such as glucose starvation (*Moruno-Manchon et al., 2013*), interferon-γ stimulation (*Matsuzawa et al., 2012*) reservatrol (*Chang et al., 2014*) or accumulation of mutant gilal fibrillary acidic protein in astrocytes (*Tang et al., 2008*). However, Webber and Tooze found that p38α functions in the negative regulation of basal and starvation-induced autophagy in HEK293 cells (*Webber and Tooze, 2010*). The basis for the apparent discrepancy between this previous report and our current findings may reflect that p38α functions differently in different cell types or may reflect other differences in experimental design. For example, we did not directly examine the effects of p38α overexpression or p38α siRNA knockdown on MK2/MK3 activation and autophagy; instead, we used a dominant-negative p38α plasmid. One possibility is that there may be compensatory activation of different p38 isoforms that have different substrate signaling specificity in different experiments. In support of the concept that different p38 isoforms have different effects on starvation-induced MK2/MK3 activation, we found that dominant-negative p38β, but not dominant-negative p38γ, also blocks starvation-induced MK2/MK3 activation (unpublished data). Another possibility is that p38α may function upstream of MK2/MK3 in amino acid starvation, but its effects on other substrates may, in some contexts, counterbalance such effects in terms of autophagy regulation. Given our findings that dominant negative p38α blocks starvation-induced MK2/MK3 activation, further studies are warranted to more clearly define the role of this signaling event in amino acid starvation-induced autophagy.

## Beclin 1 S90 phosphorylation is essential for its tumor suppression function

*Beclin 1* is a haploinsufficient tumor suppressor; it is frequently monoallelically deleted in human breast and ovarian cancers (*Aita et al., 1999*); heterozygous loss in mice results in an increased incidence of breast and other spontaneous tumors (*Qu et al., 2003*; *Yue et al., 2003*; *Cicchini et al., 2014*); decreased *beclin 1* mRNA expression is associated with aggressive clinic-pathological features and poor prognosis in human breast cancer (*Tang et al., 2015*); and *beclin 1* gene transfer impairs the ability of MCF7 human breast tumor cells to form tumor xenografts in immunodeficient mice (*Liang et al., 1999*). Previous studies have shown that mutation of the nuclear export signal of Beclin 1 (resulting in its nuclear retention) or deletion of the evolutionary conserved domain of Beclin 1 (which abrogates it binding to VPS34 and autophagy function) blocks its tumor suppressor activity (*Liang et al., 2001*; *Furuya et al., 2005*). Our observation that a single point mutation in Beclin 1 that abrogates starvation-induced autophagy, Beclin 1 S90A, also eliminates it ability to suppress MCF7 mammary tumorigenesis provides additional support for the concept that Beclin 1 functions as a tumor suppressor through its autophagy activity. However, as this mutation impairs Beclin 1-associated VPS34 lipid kinase activity, we cannot rule out the possibility that the mutation blocks a VPS34-dependent, autophagy-independent function of the Beclin 1/VPS34 complex in tumor suppression.

Regardless of the precise mechanisms by which the Beclin 1 S90A mutation blocks the tumor suppressor function of Beclin 1, our data show a strong association between decreased autophagy, increased tumor growth, and increased cellular proliferation in human mammary tumor xenografts. This association occurs despite the presence of increased cell death in tumors with decreased autophagy, and is consistent with our previous findings showing that enhanced suppression of autophagy in non-small cell lung carcinomas is associated with enhanced tumor growth despite increased cell death (*Wei et al., 2013*). Taken together, these findings suggest that the pro-survival function of autophagy in established tumors may not be the most important determinant of net tumor growth; other factors such as the effects of autophagy on cell growth control and genomic stability may be more important.

Another implication of our finding relates to the potential relationship of MK3 deletion and Beclin 1 S90 phosphorylation in cancer development. Given the requirement of the Beclin 1 S90 site for the tumor suppressor activity of Beclin 1, it will be interesting to determine whether the lack of MK3-dependent Beclin 1 S90 phosphorylation contributes to the pathogenesis of small cell lung carcinomas that commonly harbor deletions in MK3 (*Sithanandam et al., 1996*) or of other cancers with frequent heterozygous loss of MK3 such as invasive breast carcinomas, ovarian carcinomas, lung adenocarcinomas and lung squamous cell carcinomas (www.cbioportal.org).

## Beclin 1 S90 phosphorylation may stimulate autophagy by modulating the activity of the class III phosphatidylinositol 3-kinase complex I (PI3KC3–C1)

PI3KC3-C1 consists of the lipid kinase, VPS34; the scaffolding protein, VPS15; and two autophagy proteins containing coiled-coil domains, ATG14 and Beclin 1. Our data show that Beclin 1 S90 phosphorylation increases the VPS34 lipid kinase activity of this complex, without altering its composition or membrane localization. These data are consistent with predictions from a recent EM reconstitution of the ordered domains of the PI3KC3-C1 complex (*Baskaran et al., 2014*). The architecture of the EM reconstituted complex predicts that signaling inputs at the N-terminus of Beclin 1 are likely to function in the allosteric regulation of its lipid kinase activity. There are no direct interactions between ATG14 and Beclin 1 and VPS34; however, the unstructured N-terminal region of Beclin 1 is predicted to be in proximity to the catalytic subunit of VPS34, leading to the proposal that signals in the region of the Beclin 1 N-terminus regulate the activity of VPS34 lipid kinase domain. We speculate that MK2/MK3-dependent phosphorylation of Beclin 1 S90 may function to initiate autophagy precisely through the mechanism predicted by the recently solved structure of the PI3KC3-C1 complex, that is, through allosteric interactions that increase the activity of the VPS34 lipid kinase domain.

## BCL2 inhibits autophagy through a mechanism involving suppression of MK2/MK3-dependent Beclin 1 S90 phosphorylation

Beclin 1 was originally identified as a BCL2-interacting protein (*Liang et al., 1998*), and BCL2 and BCL2L1 inhibit autophagy by binding to the BH3 domain of Beclin 1 (*Pattingre et al., 2005*; *Maiuri et al., 2007*). Bcl2 plays a crucial role in the regulation of stimulus-induced autophagy in vivo, as mice that contain non-phosphorylatable mutations in Bcl2 that prevent disruption of its binding to Beclin 1 are deficient in both starvation- and exercise-induced induced autophagy (*He et al., 2012*). Despite the importance of BCL2/BCL2L1 as negative regulators of autophagy, little is understood about the precise molecular mechanism by which their binding to Beclin 1 inhibits its autophagy function. The BH3 domain is in the N-terminal half of Beclin 1 and is not required for interaction with ATG14 or VPS34 or for the autophagy function of Beclin 1. Homodimerization of Beclin 1 favors binding BCL2/BCL2L1 and disfavors binding ATG14 or UVRAG (*Noble et al., 2008*), but the temporal order of Beclin 1 homodimerization and binding BCL2/BCL2L1 are not known. Even if BCL2/BCL2L1 binding to Beclin 1 precedes its homodimerization, it is not yet established that the promotion of Beclin 1 homodimerization and consequent prevention of binding to ATG14 is an important mechanism by which these proteins inhibit the autophagy function of Beclin 1.

Our data uncover a previously undescribed mechanism by which BCL2 inhibits the autophagy function of Beclin 1. BCL2 binding to the BH3 domain of Beclin 1 blocks an essential step in the positive regulation of starvation-induced autophagy—MK2/MK3-dependent Beclin 1 S90 phosphorylation. Our structure-based model predicts that this may occur via steric hindrance of MK2/MK3 binding to

Beclin 1 when BCL2 is bound to the BH3 domain of Beclin 1, which is in close proximity to Beclin 1 serine 90. While further studies will be required to test this structural model, our data provide strong functional evidence that BCL2 binding to Beclin 1 blocks Beclin 1 S90 phosphorylation, both in vitro and in vivo. BCL2 inhibits MK2-dependent in vitro phosphorylation of Beclin 1 S90 in a dose-dependent manner; BCL2 binding to Beclin 1 in cultured cells prevents starvation and active MK2-dependent Beclin 1 S90 phosphorylation; and Bcl2 AAA mice have deficient starvation-induced Beclin 1 S90 phosphorylation despite starvation-induced activation (phosphorylation) of MK2.

## MAPK signaling pathway molecules converge on Beclin 1 activation in starvation-induced autophagy

Taken together, our findings describe a central regulatory loop underlying starvation-induced activation of autophagy. This loop involves two distinct arms of the MAPK signaling pathway, which act in parallel to activate the autophagy function of Beclin 1 in response to starvation (*Figure 8*). Previously defined JNK1-mediated BCL2 multisite phosphorylation, leading to disruption of BCL2/Beclin 1 binding (*Wei et al., 2008*), is a requisite event for subsequent for MK2/MK3-dependent Beclin 1 S90 phosphorylation and Beclin 1-dependent autophagy. The dual functions of two arms of the MAPK signaling pathway, JNK1 and MK2/MK3, in mediating starvation-induced autophagy at the level of Beclin 1 S90 phosphorylation underscore the crucial importance of the interaction between the MAPK signaling pathway and Beclin 1 in starvation-induced autophagy. More broadly, these findings suggest that activation of autophagy is intricately coordinated with other MAPK signaling stress responses.

## Materials and methods

### Cell culture

HeLa, HEK293T, and MCF7 cells and SV40 immortalized murine embryonic fibroblasts (MEFs) retrovirally transduced with pMMP-IRES empty vector or pMMP-IRES MK2 (*Ronkina et al., 2011*) were grown in DMEM supplemented with 10% FBS, 100 U/ml penicillin and 100 µg/ml streptomycin. Doxycycline-inducible *beclin 1* shRNA U2OS cells (*Sun et al., 2008*) were grown in DMEM supplemented with 10% tetracycline-free FBS. HeLa/Control, HeLa/BCL2, HeLa/BCL2 AAA, and SKNSH/3xFlag-Beclin 1 cells were generated by transfection of the parental cells with pIRES.Neo, pIRES.Neo/Myc-BCL2, pIRES.Neo/Myc-BCL2 AAA or pBICEP-CMV2-Beclin 1, and selection with 0.5 µg/ml G418. MCF7 cells retrovirally transduced with pBABE-puro empty vector or Flag-Beclin 1 (WT or S90A) were grown in DMEM supplemented with 10% FBS, 0.01 mg/ml human recombinant insulin and 1 µg/ml puromycin. $MK2^{-/-}/MK3^{-/-}$ MEFs retrovirally transduced with pBABE-puro empty vector or Flag-Beclin 1 (WT or S90 mutants) were grown in DMEM supplemented with 10% FBS and 1 µg/ml puromycin. For autophagy assays, cells were grown in either the DMEM media described above for each cell line supplemented with 2× non-essential amino acids and 2 mM glutamine (nutrient-rich conditions) or in HBSS (starvation conditions) for the indicated time period. The HBSS used in starvation conditions contained glucose.

### Plasmids

The human *beclin 1* gene was inserted into the EcoRI/BamH1 restriction site of the pBICEP-CMV2 vector. pBICEP-CMV2-Beclin 1 S90 mutants and pBABE-puro-Beclin 1 S90 mutants were constructed using a QuickChange site mutagenesis kit (Agilent Technologies, Santa Clara, CA). The shRNA non-targetable Beclin 1 constructs harbor silent mutations (mutated sequence: GGAAGCAAAACTAG TAACA) in the region of *beclin 1* targeted by shRNA (GGGTCTAAGACGTCCAACA) (sense strand) in order to rescue the effects of knockdown of endogenous Beclin 1. All of the constructs were confirmed by sequencing. Other plasmids used in this study have been previously described, and include plasmids expressing GFP-LC3 (*Mizushima et al., 2004*), dominant-negative MK2 (MK2K76R) (*Winzen et al., 1999*), constitutively active MK2 (T205E/T317E) (*Engel et al., 1995*), dominant-negative Flag-p38α (Addgene plasmid #20352 [*Enslen et al., 1998*]), and dominant-negative JNK1 (*Lei et al., 2002*).

### Antibodies

Flag was detected using an anti-Flag M2 HRP antibody (1:2000) (Sigma-Aldrich, St. Louis, MO). The phosphospecific Beclin 1 S90 antibody was produced by PhosphoSolutions (Aurora, CO). Briefly,

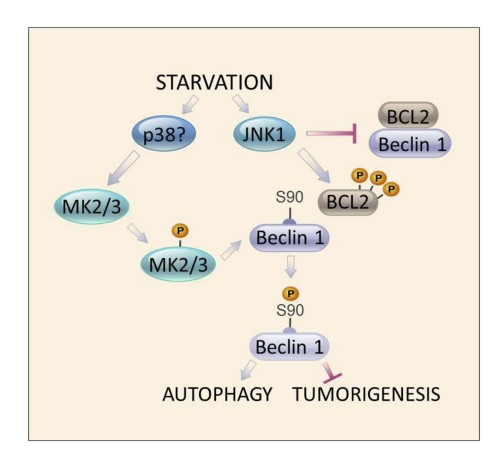

**Figure 8**. Model of convergent functions of MAPK signaling molecules in amino acid starvation-induced autophagy and regulation of Beclin 1 S90 phosphorylation. In response to nutrient starvation, JNK1 results in multi-site phosphorylation of BCL2 and disruption of BCL2/Beclin 1 binding (**Wei et al., 2008**). This permits active MK2/3 to phosphorylate Beclin 1 at residue serine 90 (**Figures 4–5**), which is essential for the autophagy and tumor suppressor function of Beclin 1 (**Figures 2–3**). Mutations in BCL2 that block multi-site phosphorylation by JNK1 (and block starvation-induced disruption of BCL2/Beclin 1 binding [**He et al., 2012**]) block MK2-mediated Beclin 1 S90 phosphorylation both in vitro and in vivo (**Figure 7**). The MAPK signaling molecule, p38, may function upstream to activate MK2/MK3 during amino acid starvation (**Figure 5—figure supplement 1**); however, a definitive role for other kinases has not been excluded. Together, our data suggest a model in which autophagy activation during amino acid starvation requires two simultaneous functions of MAPK signaling molecules: (1) MK2/MK3-mediated Beclin 1 S90 phosphorylation; and (2) JNK1-mediated disruption of BCL2 binding to Beclin 1 (which functions to block MK2/MK3-mediated Beclin 1 S90 phosphorylation).

synthetic peptides corresponding to phosphorylated and dephosphorylated S90 of Beclin 1 were injected to two rabbits and sera were purified using a phosphopeptide affinity column followed by a dephosphopeptide affinity column. The phosphospecific Beclin 1 S90 antibody was used at a concentration of 1:500. Beclin 1, HSP27, p-HSP27, Actin, MK2, p62, LC3, TOM20, PDI, and GAPDH were detected using a rabbit or goat anti-Beclin 1 antibody (Santa Cruz Biotechnology, Dallas, TX, 1:1000 dilution), a rabbit anti-HSP27 antibody (Santa Cruz Biotechnology, 1:200 dilution), a rabbit anti-p-HSP27 antibody (Santa Cruz Biotechnology, 1:200 dilution), an anti-β-Actin HRP antibody (Santa Cruz Biotechnology, 1:2000 dilution), a rabbit anti-MK2 antibody (Cell Signaling Technology, Beverly, MA; 1:1000 dilution), a mouse anti-p62 antibody (Abnova, Walnut, CA; 1:2000 dilution) a rabbit anti-LC3 antibody (Novus Biologicals, Littleton, CO; 1:1000 dilution), a rabbit anti-TOM20 antibody (Santa Cruz Biotechnology; 1:1000 dilution), a rabbit anti-PDI antibody (Cell Signaling Techology; 1:1000 dilution), and a mouse anti-GAPDH antibody (Chemicon International, Temecula, CA; 1:000 dilution).

## Transient transfection

Lipofectamine 2000 (Life Technologies, Grand Island, NY) was used for cell transfection according to the manufacturer's protocol. For the GFP-LC3 assay, GFP-LC3 and Beclin 1 (or Beclin 1 S90 mutants) were co-transfected at a molecular ratio of 1:2. For western blot analyses in *beclin 1* shRNA U2OS cells, shRNA non-targetable Beclin 1 (or Beclin 1 S90 mutants) and MK2 mutants were co-transfected at a molecular ratio of 1:1.

## Protein extraction and immunoblotting

Cells were lysed in lysis buffer (100 mM Tris pH 8.0, 150 mM NaCl, 1 mM EDTA, 1% Triton X-100) with protease inhibitors (Roche Life Sciences, Indianapolis, IN) and Pierce phosphatase inhibitors (Thermo Fisher Scientific, Rockford, IL) for 30 min. Cell lysates were separated on 4–15% TGX gels (Bio-Rad Laboratories, Hercules, CA), transferred to PVDF membranes (Bio-Rad Laboratories), and probed with the indicated antibodies. For immunoblotting levels of proteins in subcellular fractions, cytsolic, mitochondrial, and microsomal fractions were separated with a Qiagen Qproteome Mitochondrial Isolation Kit (Qiagen, Valencia, CA), and resuspended in an equal volume of storage buffer.

## Radioactive isotope labeling

HeLa cells were plated on 6 cm dishes, and transfected the following day with pBICEP-CMV2, pBICEP-CMV2-Beclin 1, or pBICEP-CMV2-Beclin 1 S90A using lipofectamine 2000. On the second day after transfection, cells were grown in phosphate-free medium (phosphate free DMEM + 10% FBS) for 3 hr. The cells were then washed with phosphate-free EBSS three times and were grown in phosphate-free EBSS or phosphate-free normal medium supplemented with 40 μCi $^{33}$P-orthophosphate for 2 hr.

The cells were washed three times with PBS, and lysed with lysis buffer (100 mM Tris pH 8.0, 150 mM NaCl, 1 mM EDTA, 1% Triton X-100 with protease and phosphatase inhibitors) for 30 min. Lysates were centrifuged for 10 min at 13,000 rpm at 4°C. The supernatants were collected and adjusted to equal protein concentrations using the Bradford assay. 9 µl α-Flag (M2) agarose (Sigma-Aldrich) was added to each sample and samples were rotated at 4°C overnight. The agarose was washed three times with PBS. 30 µl SDS loading buffer was added to each sample and the samples were boiled at 95°C for 5 min. 10 µl of each sample were loaded on 4–15% TGX gels (Biorad Laboratories).

## Peptide synthesis and kinase screen

Peptides based on amino acid sequences of human Beclin 1 83–97 were synthesized (JPT Peptide Technologies, Berlin, Germany) and used as substrates for an in vitro protein kinase screen. The peptides were dissolved in 50 mM HEPES, pH 7.5 at a concentration of 200 µM. A radiometric Streptavidin-FlashPlate-based protein kinase assay was performed to measure the kinase activity of 190 serine/threonine kinases (ProQinase, Freiburg, Germany). For mutational analysis, peptides (wild-type and S90A) based on amino acid sequences of human Beclin 1 83–103 were used.

## Immunoprecipitation

Immunoprecipitation of endogenous Beclin 1 was performed using a polyclonal goat anti-Beclin 1 antibody (Santa Cruz Biotechnology; 1:50 dilution) and immunoprecipitation of Flag-tagged Beclin 1 was performed using a monoclonal anti-Flag M2 antibody pre-conjugated to agarose (Sigma–Aldrich; 1:20 dilution). For co-immunoprecipitation of Beclin 1 and BCL2, immunoprecipitation was performed using a polyclonal rabbit anti-Beclin 1 antibody (Santa Cruz Biotechnology; 1:50 dilution) and BCL2 was probed with a monoclonal HRP antibody (Santa Cruz Biotechnology; 1:200 dilution).

## MK2/3 and VPS34 in vitro kinase assays

For MK2/3 in vitro kinase assays, HEK293T cells were transfected with pBICEP-CMV2-Beclin 1 or pBICEP-CMV2-Beclin 1 S90A using lipofectamine 2000 (Invitrogen; Grand Island, NY); 6 µg DNA was transfected into each 10 cm dish. After 24 hr, each dish of cells was lysed with 500 µl lysis buffer (100 mM Tris pH 8.0, 150 mM NaCl, 1 mM EDTA, 1% Triton X-100 with protease inhibitors) for 30 min at 4°C. Flag-Beclin 1 was immunoprecipitated using 75 µl anti-Flag (M2) agarose (Sigma-Aldrich) for 3 hr at 4°C. The beads were washed twice with lysis buffer and another two times with phosphatase buffer. The samples were then treated with λ-protein phosphatase (New England Biolabs, Ipswich, MA) according to the manufacturer's protocol. For 60 µl Flag-agarose, 4 µl λ-protein phosphatase was used. The agarose was washed twice with lysis buffer with phosphatase inhibitors, and another two times with kinase reaction buffer (60 mM HEPES, pH 7.6, 3 mM $MgCl_2$, 3 mM $MnCl_2$, 3 mM $Na_3VO_4$, 1.2 mM DTT). For each in vitro kinase reaction, 15 µl Flag-agarose was mixed with 15 µl kinase buffer (with 1 µM ATP). The kinases were added to a final concentration of 0.16 µM. The reactions were incubated at 30°C for 30 min. 30 µl SDS loading buffer was added to each reaction and the samples were incubated at 95°C for 2 min. After brief centrifugation, the supernatants were loaded onto 4–15% TGX gels (Bio-Rad Laboratories), transferred to PVDF membranes and incubated with the indicated antibodies. To assess whether BCL2 blocks MK2-mediated phosphorylation of Beclin 1 in vitro, 5 µg GST (negative control) or various amounts of GST-BCL2 recombinant protein were added to each reaction tube.

To measure Beclin 1-associated VPS34 in vitro lipid kinase activity, Beclin 1-VPS34 complexes were immunoprecipitated from U2OS cells with anti-Flag and used as substrates for a VPS34 in vitro lipid kinase assay as described (*Wang et al., 2012*; *Wei et al., 2013*). The thin layer chromatography spots and western blot band intensities were measured by densitometry (Visionworks LS image acquisition and analysis software, UVP LLC).

## Tumor xenograft studies

A 60-day slow-release 1.7 mg estrogen pellet (Innovative Research of America, Sarasota, Florida) was injected in the neck region of 7-week old female *nu/nu* mice. After 1 week, 5 × 10^6 MCF7 cells retrovirally transduced with Beclin 1 (WT or S90 mutants) were injected into the upper right mammary fat pad of each mouse. Tumor width and length were measured twice a week for a total experimental

duration of 8 weeks. Each group contained 11 mice. All animal experiments were performed in accordance with institutional guidelines and approved by the UT Southwestern Medical Center Institutional Animal Care and Use Committee.

## Histology

Mouse tumor xenografts were fixed in 4% PFA, embedded in paraffin, and sectioned (~5 microns in thickness). Immunohistochemical staining of paraffin-embedded tumor tissues was performed using anti-p62 (PROGEN Biotechnik, Heidelberg, Germany; 1:2000 dilution) or anti-Ki67 (Abcam, Cambridge, MA; 1:200 dilution) primary antibodies and the ABC Elite immunoperoxidase kit (Vector Laboratories, Burlingame, CA) according to the manufacturer's instructions. TUNEL staining was performed according to the manufacturer's instructions, using Sigma FAST 3, 3′-diaminobenzidine (DAB) tablets as the peroxidase substrate. Reciprocal intensity of p62 was quantified using the ImageJ 1.47v software (http://imagej.nih.gov/ij).

## Mouse starvation and tissue preparation

8-week-old Bcl2$^{AAA}$ knock-in and littermate control C57/BL6 mice (*He et al., 2012*) were starved for 48 hr and thigh muscles were snap-frozen in liquid nitrogen. The thigh muscles (vastus lateralis) were homogenized and protein was extracted in lysis buffer (100 mM Tris, pH 8.0, 150 mM NaCl, 1 mM EDTA, 1% Triton X-100) with protease (Roche Life Sciences) and Pierce phosphatase inhibitors cocktail (Thermo Fisher Scientific) prior to western blot analyses with indicated antibodies.

## Acknowledgements

We thank Haley Harrington for assistant with manuscript preparation; Lori Nguyen for technical assistance; and Noboru Mizushima and Roger Davis for providing critical reagents. This work was supported by National Institutes of Health grants RO1 CA84254 (BL), RO1 CA109618 (BL), K08 AI099150 (RS), and RO3 NS090939 (SS); a Cancer Prevention and Research Institute of Texas grant RP120718-PI (BL); an NSF grant MCB-1413525 (SS); an ND EPSCoR project FAR0023534 (SS); and grant SFB 566 B12 from Deutsche Forschungsgemeinschaft (MG).

## Additional information

### Competing interests

BL: Reviewing editor, *eLife*. The other authors declare that no competing interests exist.

### Funding

| Funder | Grant reference number | Author |
|---|---|---|
| Howard Hughes Medical Institute (HHMI) | | Beth Levine |
| National Institutes of Health (NIH) | RO1 CA84254 | Beth Levine |
| Cancer Prevention and Research Institute of Texas (CPRIT) | RO1 RP120718 | Beth Levine |
| National Institutes of Health (NIH) | CA109618 | Beth Levine |
| National Institutes of Health (NIH) | K08 AI1099150 | Rhea Sumpter Jr |
| National Institutes of Health (NIH) | RO3 NS090939 | Sangita Sinha |
| National Science Foundation (NSF) | MCB-1413525 | Sangita Sinha |
| Deutsche Forschungsgemeinschaft | SFB 566 B12 | Matthias Gaestel |

The funders had no role in study design, data collection and interpretation, or the decision to submit the work for publication.

### Author contributions

YW, ZA, Conception and design, Acquisition of data, Analysis and interpretation of data, Drafting or revising the article; ZZ, MS, Acquisition of data, Analysis and interpretation of data; RS, Acquisition

of data, Analysis and interpretation of data, Drafting or revising the article; XZ, SS, MG, BL, Conception and design, Analysis and interpretation of data, Drafting or revising the article

## Ethics

Animal experimentation: This study was performed in strict accordance with the recommendations in the Guide for the Care and Use of Laboratory Animals of the National Institutes of Health. All of the animals were handled according to approved institutional animal care and use committee (IACUC) protocols (2012-0193) of the University of Texas Southwestern Medical Center.

## Additional files

### Supplementary file

• Supplementary file 1. Raw data of in vitro kinase screen with 190 kinases using the Beclin 1 peptide spanning from amino acids 83–97 as a substrate.

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
