## [Decision Letter]

Thank you for sending your work entitled “Stress-Responsive MAPK Family Members, MK2/MK3, Activate Starvation-Induced Autophagy through Beclin 1 Phosphorylation” for consideration at *eLife*. Your article has been favorably evaluated by Vivek Malhotra (Senior editor) and three reviewers. Vivek Malhotra also acted as Reviewing editor.

The Reviewing editor and the reviewers discussed their comments before we reached this decision, and the Reviewing editor has assembled the following comments to help you prepare a revised submission.

The reviewers agreed on the significance of the findings and the high quality of the data presented. But how starvation activated MK2/MK3 and how activation of Beclin2 by MK2 initiated autophagy was not clear from the data presented. It was generally agreed that addressing some aspect of this would strengthen the conclusions.

We suggest that you address the following three major issues:

1) How activation of Beclin 1 by MK2 results in the initiation of autophagy? Results shown in Figure 2 suggest a possible model, as cells expressing S90A display reduced VPS34 lipid kinase activity. However, additional mechanistic insight is needed. Might Beclin 1 S90 phosphorylation stabilize the Beclin 1/VPS34/ATG14 complex? The authors should blot for VPS34 in their immuno-precipitation experiments to monitor the interaction of VPS34 with wild type Beclin 1 and the S90A and S90E mutant proteins. In an alternative model, does S90 phosphorylation stabilize interaction of the Beclin 1/VPS34/ATG14 complex with membranes, similar to a previously suggested model for ATG14 regulation (14)? The authors should monitor membrane binding of wild type Beclin 1 and the S90A and S90E mutant proteins using cellular fractionation approaches and in vivo localization experiments. In yet another possible model, might Beclin 1 S90 phosphorylation induce autophagy by stimulating VPS34 lipid kinase activity, similar to ULK1-mediated phosphorylation of Beclin 1 (55)?

2) MK2 and MK3 have not been previously implicated in amino acid starvation stress responses. Moreover, the best-characterized kinase that activates MK2/MK3, p38 MAPK, is thought to negatively regulate starvation-induced autophagy (Weber and Tooze, 2010). It would be very interesting and important to provide some new insight into how amino acid starvation results in MK2/MK3 activation.

3) Why bafilomycin A1 treatment alone induces Beclin1 phosphorylation as seen in Figure 2?

---

## [Author Response]

*The reviewers agreed on the significance of the findings and the high quality of the data presented. But how starvation activated MK2/MK3 and how activation of Beclin2 by MK2 initiated autophagy was not clear from the data presented. It was generally agreed that addressing some aspect of this would strengthen the conclusions*.

We are delighted that the reviewers agreed on the significance of our findings and the high quality of the data presented. In the revised manuscript, we have added additional experiments to address how starvation activates MK2/MK3. In addition, we have performed the experiments suggested by the reviewers to further address how MK2-mediated phosphorylation of Beclin 1 initiates autophagy. We thank the reviewers for their suggestions, and believe these additional experiments significantly strengthen our manuscript.

*We suggest that you address the following three major issues*:

*1) How activation of Beclin 1 by MK2 results in the initiation of autophagy? Results shown in*
Figure 2
*suggest a possible model, as cells expressing S90A display reduced VPS34 lipid kinase activity. However, additional mechanistic insight is needed. Might Beclin 1 S90 phosphorylation stabilize the Beclin 1/VPS34/ATG14 complex? The authors should blot for VPS34 in their immuno-precipitation experiments to monitor the interaction of VPS34 with wild type Beclin 1 and the S90A and S90E mutant proteins*.

As requested, we have performed immunoblots for VPS34 (and also ATG14) in the Flag-Beclin 1 immunoprecipitates in Figure 2. The results (added to revised Figure 2) indicate that there are no differences in the Beclin 1/VPS34/ATG14 complex related to the phosphorylation status of Beclin 1, i.e. equivalent amounts of VPS34 and ATG14 co-immunoprecipitate with wild-type Beclin 1, the Beclin 1 S90A phosphorylation -defective mutant, and the Beclin 1 S90E phosphomimetic mutant, and the amounts do not change in response to starvation. These data indicate that Beclin 1 S90 phosphorylation increases VPS34 lipid kinase activity without stabilizing the Beclin 1/VPS34/ATG14 complex.

*In an alternative model, does S90 phosphorylation stabilize interaction of the Beclin 1/VPS34/ATG14 complex with membranes, similar to a previously suggested model for ATG14 regulation (*[14]*)? The authors should monitor membrane binding of wild type Beclin 1 and the S90A and S90E mutant proteins using cellular fractionation approaches and in vivo localization experiments*.

As requested by the reviewers, we have performed cellular fractionation experiments to examine the membrane localization of wild-type Beclin 1, Beclin 1 S90A, and Beclin 1 S90E, as well as ATG14 and VPS34 in basal and starvation conditions. These results show that the amounts of the different Beclin 1 constructs in each subcellular fraction (cytosol, mitochondrial-enriched fraction, microsomal fraction) are similar and do not vary in response to starvation. We observed similar findings for ATG14. For VPS34, although there are some differences in the different lanes, we did not observe any consistent pattern in several repeat experiments with respect to amounts of VPS34 in the microsomal fraction during starvation or in cells expressing the different Beclin 1 constructs. Thus, we do not believe that Beclin 1 S90 phosphorylation increases the membrane binding of the Beclin 1/VPS34/ATG14 complex. These new experiments are shown in Figure 2—figure supplement 3 of the revised manuscript. Also, in earlier immunostaining experiments (data not shown) performed by our laboratory, we did not find any difference in the subcellular localization patterns of wild-type Beclin 1, Beclin 1 S90A, and Beclin 1 S90E.

*In yet another possible model, might Beclin 1 S90 phosphorylation induce autophagy by stimulating VPS34 lipid kinase activity, similar to ULK1-mediated phosphorylation of Beclin 1 (*[55]*)*?

Our results in Figure 2 are indeed consistent with a model in which Beclin 1 S90 phosphorylation stimulates VPS34 lipid kinase activity. To strengthen this claim, we have added to Figure 2 the quantification of multiple different experiments examining the VPS34 lipid kinase activity associated with wild-type Beclin 1, Beclin S90A, and Beclin 1 S90E (Figure 2 of revised manuscript). These results clearly show that, despite similar levels of VPS34 in immunoprecipitates, there is a significant defect in Beclin 1-associated VPS34 kinase activity during starvation when Beclin 1 is lacking the S90 phosphorylation site. In addition, there is a significant increase in baseline as well as starvation-induced Beclin 1-associated VPS34 kinase activity in cells expressing the Beclin 1 S90E phosphomimetic mutant. Of note, we have added a new section to the Discussion section to highlight that we believe the mechanism by which Beclin 1 S90 phosphorylation induces autophagy is by stimulating VPS34 lipid kinase activity. This observation is particularly timely in view of the recent *eLife* publication by Baskaran et al. (2015) which described the EM reconstitution of the ordered domains of the Beclin 1/ATG15/VPS15/VPS34 complex, and predicted that signaling input from the disordered N-terminal region of Beclin 1 was likely to result in allosteric regulation of the VPS34 lipid kinase domain.

*2) MK2 and MK3 have not been previously implicated in amino acid starvation stress responses. Moreover, the best-characterized kinase that activates MK2/MK3, p38 MAPK, is thought to negatively regulate starvation-induced autophagy (Weber and Tooze, 2010). It would be very interesting and important to provide some new insight into how amino acid starvation results in MK2/MK3 activation*.

It is true that MK2 and MK3 have not been previously implicated in amino acid starvation stress responses, and indeed, this observation represents a novel element of our findings. The best-characterized kinase that activates MK2/MK3 is, as stated, p38 MAPK. There have also been some claims that MK2/MK3 can be activated by JNK. For these reasons, we evaluated the effects of both dominant-negative p38α and dominant-negative JNK1 on starvation-induced MK2 activation (as measured by western blot detection of an MK2 band shift and phosphorylation of the downstream MK2 substrate, HSP27), as well as on Beclin 1 S90 phosphorylation. Our results, shown in Figure 5—figure supplement 1, indicate that dominant-negative p38α inhibits starvation-induced MK2 activation, Beclin 1 S90 phosphorylation, and autophagy. Dominant-negative JNK1 also inhibits starvation-induced Beclin 1 S90 phosphorylation and autophagy (presumably due to its inhibition of Bcl–2/Beclin 1 disruption as reported previously ([67], Molecular Cell) and our findings in this manuscript showing that Bcl–2 binding to Beclin 1 blocks Beclin 1 S90 phosphorylation); however, in contrast to dominant-negative p38α, dominant-negative JNK1 does not block starvation-induced MK2 activation. Thus, these results suggest that p38α, not JNK1, is indeed the kinase that functions upstream of MK2/MK3 in amino acid starvation-induced Beclin 1 S90 phosphorylation. To further confirm that MK2 functions downstream of p38 in amino acid starvation, we also evaluated whether constitutively active MK2 could reverse the inhibitory effects of dominant negative p38α on Beclin 1 S90 phosphorylation. Our results, shown in Figure 5—figure supplement 1, indicate that constitutively active MK2 blocks the inhibitory effects of dominant negative MK2 on starvation-induced Beclin 1 S90 phosphorylation. Taken together, these results suggest that p38α MAPK functions upstream to activate MK2-dependent Beclin 1 S90 phosphorylation and autophagy during amino acid starvation.

We recognize that this finding is in apparent contradiction to the report of Webber and Tooze (2010, EMBO) that p38α negatively regulates starvation-induced autophagy. We also note that other studies report a positive role for p38α MAPK in autophagy induction, including in response to glucose starvation (Moruno-Manchón et al., 2013, Biochem J), IFNγ stimulation ([38], J Immunol; Matsuzawa et al., 2014, Immunology), reservatrol treatment ([7], J Cell Physiol), and accumulation of a mutant glial acidic fibrillary protein ([62], Hum Mol Genet). In the revised Discussion, we have discussed possible reasons for apparently discrepant findings in different studies, as well as cautioned that there is a need for further studies of p38α MAPK-regulation of starvation-induced autophagy. Unfortunately, within the scope of the present paper, it is not possible to thoroughly examine this issue and resolve the basis for discrepant reports in the literature.

*3) Why bafilomycin A1 treatment alone induces Beclin1 phosphorylation as seen in*
Figure 2?

While this manuscript focuses on starvation-induced autophagy and starvation-induced Beclin 1 S90 phosphorylation, we have also found that Beclin 1 undergoes S90 phosphorylation in response to several other autophagy-inducing stress stimuli. Therefore, we postulated that bafilomycin A1 might induce Beclin 1 S90 phosphorylation as a consequence of generation of reactive oxygen species (ROS), presumably as a consequence of its inhibition of vacuolar-type H(+)-ATPases ([70], J Toxicol Sci; [73], Cell Mol Life Sci). To address whether bafilomycin A1-mediated ROS generation was responsible for Beclin 1 S90 phosphorylation, we evaluated whether this event was blocked by concurrent treatment with an ROS scavenger, N-acetyl-L-cysteine (NAC). Our results indicate that bafilomycin A1 induces S90 phosphorylation of endogenous Beclin 1 in HeLa cells and that this is blocked by pretreatment with NAC. These results are shown in Figure 2—figure supplement 2 of the revised manuscript. Of note, we performed these experiments in HeLa cells to confirm that bafilomycin A1-induced Beclin 1 S90 phosphorylation occurs on endogenous Beclin 1, as the experiments in Figure 2 and Figure 2 involve the measurement of S90 phosphorylation of transfected Beclin 1 (in MCF7 cells and U2OS) and we wanted to exclude the possibility that bafilomycin A1-induced S90 phosphorylation was an artifact of exogenous Beclin 1 expression.